# NuwaTS: a Foundation Model Mending Every Incomplete Time Series

## Abstract

Time series imputation is critical for many real-world applications and has been widely studied. However, existing models often require specialized designs tailored to specific missing patterns, variables, or domains which limits their generalizability. In addition, current evaluation frameworks primarily focus on domain-specific tasks and often rely on time-wise train/validation/test data splits, which fail to rigorously assess a model's ability to generalize across unseen variables or domains. In this paper, we present **NuwaTS**, a novel framework that repurposes Pre-trained Language Models (PLMs) for general time series imputation. Once trained, NuwaTS can be applied to impute missing data across any domain. We introduce specialized embeddings for each sub-series patch, capturing information about the patch, its missing data patterns, and its statistical characteristics. By combining contrastive learning with the imputation task, we train PLMs to create a versatile, one-for-all imputation model. Additionally, we employ a plug-and-play fine-tuning approach, enabling efficient adaptation to domain-specific tasks with minimal adjustments. To evaluate cross-variable and cross-domain generalization, we propose a new benchmarking protocol that partitions the datasets along the variable dimension. Experimental results on over seventeen million time series samples from diverse domains demonstrate that NuwaTS outperforms state-of-the-art domain-specific models across various datasets under the proposed benchmarking protocol. Furthermore, we show that NuwaTS generalizes to other time series tasks, such as forecasting.

## 1 Introduction

Time series data are pervasive across numerous fields, including transportation (Li et al., 2018), healthcare (Tonekaboni et al., 2020), education (Mao et al., 2024), and meteorology (Bi et al., 2023). However, real-world time series data often suffer from missing values. Incomplete data complicate various time series applications such as forecasting and classification, ultimately degrading the quality of data analysis (Ma et al., 2020). This makes time series imputation especially critical (Wang et al., 2024a). Traditionally, time series imputation methods have leaned heavily on statistical techniques like mean imputation and interpolation (Van Buuren & Groothuis-Oudshoorn, 2011). Yet, with the rise of deep learning, there has been a notable shift towards deep learning models for imputation tasks (Cao et al., 2018; Du et al., 2023).

In traditional deep learning-based imputation models, the typical protocol involves training a model on relatively complete or incomplete time series data within a closed domain from historical records. During the testing phase, the model is validated or tested on newly observed incomplete data, which are future observations of the variables from the training set (Cao et al., 2018; Wu et al., 2022) (Figure 1(b)). However, this approach presents a key limitation: **the lack of cross-variable generalization capability**. Models trained in this manner may struggle to extend to variables not observed in the training set. For instance, in one particular factory, a newly launched production line generates several new time series data, and it's uncertain whether the model trained on existing production lines can generalize effectively to those new ones. A more challenging scenario arises when we lack training data from a specific domain and must rely on a model trained from other domains for imputation. This introduces a more complex **cross-domain generalization problem** (Jin et al., 2022; Wilson et al., 2020). For example, generalizing from traffic time series to factory time

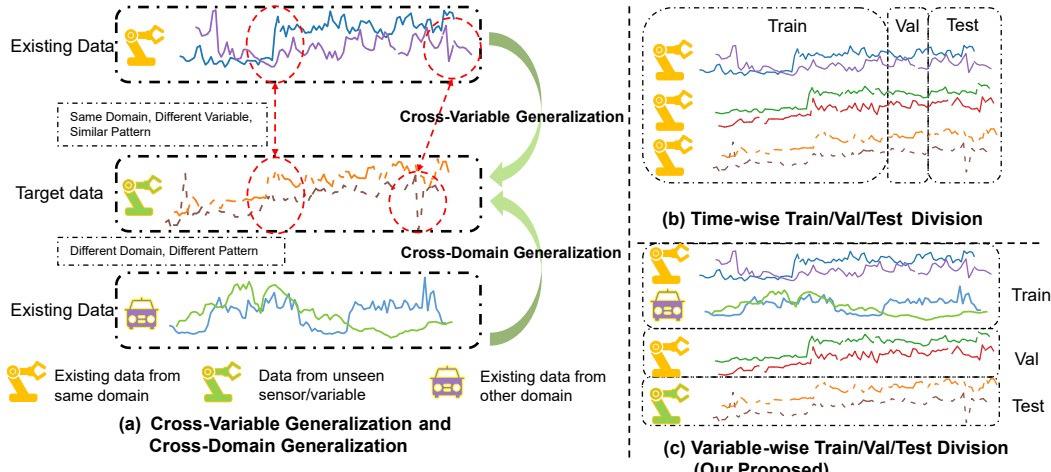

Figure 1: (a) Cross-variable and cross-domain generalization: time series data across different variables and domains may exhibit both shared and distinct patterns. (b) The conventional train/validation/test division protocol of partitioning datasets along the time dimension. (c) Variable-wise division: the proposed approach trains, validates, and tests models on distinct sets of variables, ensuring the model's ability to generalize across unseen variables during deployment.

series presents significant challenges in transferring learned knowledge across fundamentally different domains (Figure 1(a)).

In this paper, we propose a novel framework and benchmarking methodology specifically designed to address and assess time series imputation methods that are capable of both cross-variable and cross-domain generalization. We draw inspiration from the success of foundation models trained on diverse datasets spanning multiple domains (Kirillov et al., 2023; Brown et al., 2020; Peng et al., 2023). These models have demonstrated remarkable ability to generalize across tasks and domains using techniques like fine-tuning (He et al., 2022; Zhuang et al., 2020) or prompting (Jia et al., 2022; Kirillov et al., 2023). In a similar vein, we propose a method that pushes time series imputation beyond the traditional single-domain paradigm, enabling generalization both across variables and across domains.

Specifically, we introduce **NuwaTS**, a foundation model designed for incomplete time series imputation. Our model segments the time series into patches and employs novel token designs to capture both statistical information and missing data patterns. Additionally, we leverage contrastive learning combined with missing data reconstruction to train an one-for-all imputation model on time series data from diverse domains. Finally, for cases requiring domain-specific adaptation, we have developed a domain-specific prefix embedding and a plug-and-play fine-tuning mechanism. This mechanism introduces continuous prompts at each layer of the frozen pre-trained foundation model without modifying its weights, enabling efficient domain specialization.

Furthermore, we introduce a novel benchmarking paradigm for time series imputation models. Rather than the conventional time-wise train/validation/test partitioning, we partition multivariate time series data along the variable dimension, allocating different variables to the training, validation, and test sets (Figure 1(c)). This approach closely simulates real-world deployment scenarios, where models trained on one domain are required to impute missing data for entirely new variables and domains, effectively assessing the model's cross-variable and cross-domain generalization capabilities.

Our contributions are summarized as follows:

1. We propose a novel and more practically relevant benchmark which divides the multivariate time series data along the variable dimension for time series imputation, which better assesses a model's ability to generalize to new data.

2. We introduce NuwaTS, designed to handle missing data imputation tasks for any incomplete time series. NuwaTS is trained on data from diverse domains and incorporates a light-

weight "plug-and-play" fine-tuning technique that requires minimal data and computational resources, making it capable of **mending every incomplete time series**.

3. Under the proposed benchmarking protocol, the one-for-all NuwaTS consistently outperforms domain-specific state-of-the-art methods in imputation tasks across nearly all missing rates. Moreover, fine-tuned NuwaTS can be extended to time series forecasting, where its forecasting results are comparable to or even better than existing domain-specific time series forecasting models.

## 2 RELATED WORKS

### 2.1 INCOMPLETE TIME SERIES IMPUTATION

Many time series imputation models are tailored to specific missing data patterns and domains, such as randomly missing traffic time series. For instance, matrix factorization models are designed for multivariate time series imputation (Yu et al., 2016). For a dataset from a specific domain with a particular missing rate, it often requires optimizing a specialized model using a low-rank prior as the optimization target for imputation. Recent advancements in deep learning techniques have shown promising results in addressing missing data imputation. Generative models such as Generative Adversarial Networks (GANs) (Luo et al., 2019; Tashiro et al., 2021) and diffusion models are used to learn the underlying distribution of the incomplete time series. Several architectures, such as Recurrent Neural Networks (RNNs) (Cao et al., 2018; Ma et al., 2019; Liu et al., 2019; Tang et al., 2020) and attention mechanisms (Du et al., 2023), are proposed to capture temporal dependencies within incomplete time series. While achieving good performance on narrow domains and missing data patterns, these models lack versatility and generalizability to diverse domains and missing data patterns.

### 2.2 FOUNDATION MODEL FOR TIME SERIES

In recent years, foundational models for time series analysis have made notable strides, primarily leveraging pre-trained backbones from NLP and aligning modalities to extend their reasoning capabilities to time series tasks. For example, Gruver et al. (Gruver et al., 2024) discovered that by encoding time series as strings of numerical digits, LLMs can perform time series forecasting with zero-shot capabilities. GPT4TS (Zhou et al., 2024) trains time series models using pre-trained GPT weights. UniTime (Liu et al., 2024a) integrates domain-specific instructions into LLMs, enhancing their generalization across domains. Time-LLM (Jin et al., 2024) employs LLMs' reasoning by framing time series data and statistical information in textual prompts, aligning time patches with word embeddings via cross-attention mechanisms for improved zero-shot learning. Autotimes (Liu et al., 2024d) utilizes precomputed text embedding as positional embeddings and an autoregressive approach for long-term forecasting. TEST (Sun et al., 2023) uses text-prototype-aligned embeddings to enhance LLMs' reasoning in time series data. $S^2$IP-LLM (Pan et al., 2024) applies seasonal-trend decomposition to time series and uses semantic space-informed prompting to retrieve appropriate prompts from word token embeddings as prefixes. aLLM4TS (Bian et al., 2024) adapts LLMs for time series representation learning by directly converting individual patches into time series sequences. Chronos (Ansari et al., 2024) trains language models from scratch on a large collection of time series data, using scaling and quantization for tokenization. This model demonstrates strong capabilities in zero-shot probabilistic forecasting. These models primarily focus on time series forecasting and do not specifically address missing data issues or the design of embeddings for missing data patterns.

## 3 METHODOLOGY

### 3.1 PRELIMINARIES AND PROBLEM STATEMENT

**Definition 3.1.** *Incomplete Time Series: We define an incomplete time series* $\mathbf{x} = \{x_1, x_2, \ldots, x_T\} \in \mathbb{R}^T$ *from domain* $\mathcal{S}$ *as a sequence of* $T$ *observations. Each observation* $x_t$ *is associated with a timestamp* $s_t$. *In practice, an observation* $x_t$ *may not be observable due to various reasons. To represent the missing values in* $\mathbf{x}$, *we introduce a masking vector* $\mathbf{m} \in \mathbb{R}^T$

*defined by:*

$$m_t = \begin{cases} 1, & \text{if } x_t \text{ is observed} \\ 0, & \text{otherwise} \end{cases} \tag{1}$$

Suppose we have acquired $N$ time series datasets $\mathcal{D} = \{\mathbf{x}^1, \mathbf{x}^2, \ldots, \mathbf{x}^N\}$ from a diverse set of domains $\{\mathcal{S}^1, \mathcal{S}^2, \ldots, \mathcal{S}^K\}$, where $\mathbf{x}^n \in \mathbb{R}^{T_n}$ represents the $n^{th}$ time series with length $T_n$. Our objective is to utilize $\mathcal{D}$ to train an imputation model, denoted as $f_{\mathbf{\Phi}}$, characterized by parameters $\mathbf{\Phi}$. For any incomplete time series $\hat{\mathbf{x}} \in \mathbb{R}^{\hat{T}}$ accompanied by any missing pattern $\hat{\mathbf{m}} \in \mathbb{R}^{\hat{T}}$ from any domain $\hat{\mathcal{S}}$, the model $f_{\mathbf{\Phi}}$ can impute the missing values in $\hat{\mathbf{x}}$ as accurately as possible.

## 3.2 MODEL ARCHITECTURE

In this work, we initialize model parameters $\mathbf{\Phi}$ with weights from PLMs. Given the constraints of computational resources, we limit our use to the parameters of the first six layers, thereby making NuwaTS more accessible to applications with limited computational resources.

To train PLMs to adapt to various domains and missing data patterns, we have implemented several key design features for training. These include Instance Normalization & Patching, Statistical Embedding, Missing Embedding, a Domain-Specific Embedding for fine-tuning and Contrastive Learning with Variable Missing Patterns. We visualize the model architecture in Figure 2.

**Instance Normalization & Patching** Time series from different domains can vary in magnitude and distribution. To address these differences, we apply reversible instance normalization (Kim et al., 2021) to each variable before feeding it into the model, with missing values set to zero. To enhance the model's ability to recognize domain information, we use a patching technique, dividing each time series segment into non-overlapping patches. These patches are then embedded into a hidden space using a shared, learnable linear projection denoted by $\mathbf{Z}_{i,(p)} \in \mathbb{R}^{D \times N}$, enabling more effective modeling of the time series data across domains.

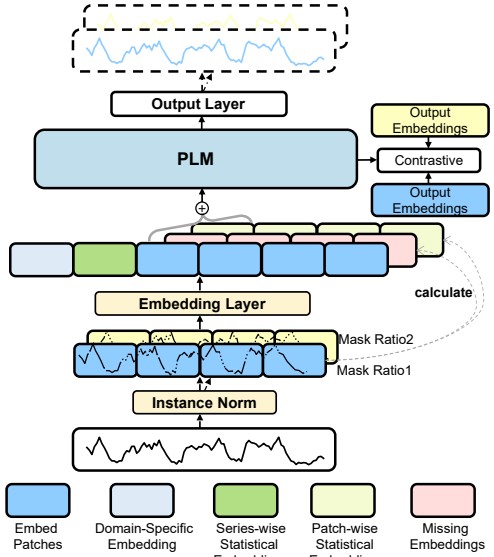

Figure 2: Overview of NuwaTS. To fully leverage the semantic information of time series and their missing patterns, NuwaTS introduces the tokenization of time series in patches. It utilizes the missing data patterns, statistical information for each pattern and the entire series, and a domain-specific embedding, trained through imputation and contrastive learning tasks.

**Statistical Embedding.** Previous work primarily used hard textual prompts to translate dataset descriptions and statistical information (Liu et al., 2024a;d; Jin et al., 2024), which proved challenging due to the mismatch between time series patches and textual prompts. In this work, we move away from this approach and instead generate statistical information, such as minimum, median, maximum values, and trends, for both the entire variable (denoted by $\mathbf{z}_{i,(v_g)} \in \mathbb{R}^D$) and individual patches (denoted by $\mathbf{Z}_{i,(v_p)} \in \mathbb{R}^{D \times N}$). This information is embedded into a hidden space using a shared, learnable linear projection, allowing for better alignment with the time series data.

**Missing Embedding.** Adapting to the missing data pattern is crucial. We design a Missing Embedding, a learnable parameter that captures the missing rate of each patch. This embedding is multiplied with the corresponding patch's mask ratio and added to the corresponding embed patches: $\mathbf{E}_{i,(p)} = \mathbf{Z}_{i,(p)} + \mathbf{Z}_{i,(v_p)} + \mathbf{z}_{i,(m)} \times \mathbf{r}_i$, allowing the model to better account for missing data across the target time series.

**Domain-Specific Embedding.** In cases where the model needs to function within a specific domain while preserving cross-domain generalization, we introduce a domain-specific embedding, $\mathbf{k} \in \mathbb{R}^D$.

This embedding learns to capture domain-specific knowledge during training and is inserted before the patch embeddings. This embedding is beneficial for the proposed fine-tuning process, as discussed in Section 3.3.

The final input embeddings are expressed as $\mathbf{E}_i = [\mathbf{k}, \mathbf{z}_{i,(v_g)}, \mathbf{E}_{i,(p)}]$, integrating both domain-specific and global statistical information. This design could help model generalize well on new in-domain data or even out-of-domain data.

**Contrastive learning.** To improve the model's adaptability to different missing patterns, we incorporate a Contrastive Learning module into the training process. For each input $\mathbf{x}_i$, we generate two random masks with different mask ratios and input the masked time series into the PLM. The module ensures the model learns similar representations for the same patch under different masks, treating them as positive samples, while treating representations from other patches and series as negative samples. We use the InfoNCE (Oord et al., 2018) loss combined with MSE loss to optimize the model (details in Appendix B.3).

**Output layer.** After passing through the PLM backbone, we discard the domain-specific and variable-wise statistical embeddings, retaining only the $N$ patch representations as inputs for the final layer. While the prefixes contribute to causal attention calculations, they are excluded from the final output. The remaining representations are flattened and linearly mapped back to their original dimensions, producing the model outputs $\mathbf{o}_i \in \mathbb{R}^L$.

### 3.3 DOMAIN-SPECIFIC FINE-TUNING

To achieve a domain-specific model, we borrow the idea of P-tuningv2 strategy (Liu et al., 2022). Unlike pre-training phase, where the domain-specific embedding $\mathbf{k}$ is only added to the input embedding.

We incorporate this domain-specific information into every layer of the PLM. Initially, we employ a domain-transfer layer—a two-layer MLP network—to map $\mathbf{k} \in \mathbb{R}^D$ to $\hat{\mathcal{K}} \in \mathbb{R}^{2 \times \text{Layer} \times D}$.

Following the P-tuningv2 approach, we also randomly initialize a continuous prompt $\mathcal{P} \in \mathbb{R}^{2 \times \text{Layer} \times D}$ for each layer. We then combine the randomly initialized $\mathcal{P}$ with the domain-specific information $\hat{\mathcal{K}}$ to serve as the domain-specific prefix Key and Value in the PLM's hidden layers. Thus, we have $[\mathbf{Key}_p, \mathbf{Value}_p] = \mathcal{P} + \beta\hat{\mathcal{K}}$, where $\beta = 0.01$. Here, $\mathbf{Key}_p$ and $\mathbf{Value}_p \in \mathbb{R}^{\text{Layer} \times D}$ contain the domain-specific prefix key and value for every layer of the pre-trained model, thereby enhancing its representational capacity.

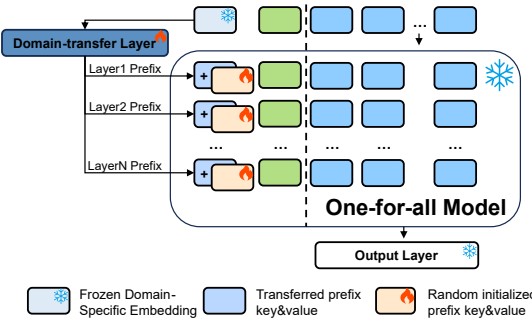

Figure 3: Illustration of domain-specific fine-tuning.

During fine-tuning on time series data from the target domain, we freeze all parameters except for the randomly initialized $\mathcal{P}$ and the domain-transfer layer. The fine-tuning process is illustrated in Figure 3. If the domain-specific prefix is removed, the domain-specific model reverts to the original one-for-all foundation model. Therefore, the fine-tuning strategy we propose is essentially "plug-and-play". Additionally, the domain-specific prefix is lightweight. For example, when using GPT-2 as the PLM backbone, the prefix requires less than 100KB of storage, compared to the 331.77MB required for the entire model. This makes our approach highly practical for real-world deployment, particularly in edge computing environments, as a single NuwaTS model can be trained on large-scale time series data. When deploying domain-specific models locally, only the corresponding prefix needs to be stored, significantly reducing storage requirements and enabling efficient deployment with minimal computational resources.

Additionally, we can incorporate inter-series correlations into the prefix. For the subsequent forecasting tasks in section 4.7, we design an inter-variable fine-tuning network to generate the layer prefix, which contains inter-series correlation information. Specific implementation details can be found in Algorithm 3 and Section B.4.

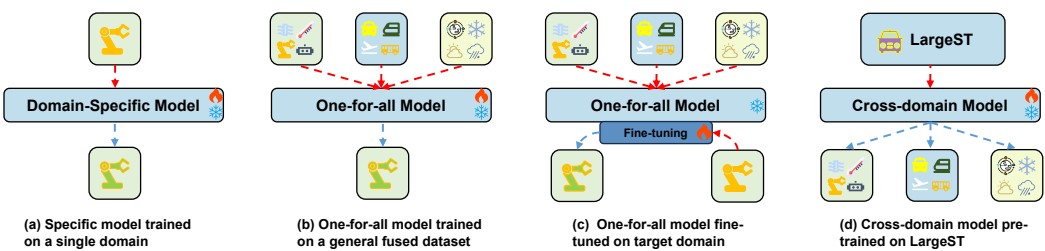

Figure 4: The four different versions of NuwaTS trained in this study.

## 4 EXPERIMENT

We conducted comprehensive experiments on NuwaTS across ten commonly used datasets from various domains. To facilitate comparison, we provided four versions of the model based on the GPT2 architecture in Figure 4: (a) a **specific model** trained on a single domain, (b) an **one-for-all model** trained on a general fused dataset (17.6M collected samples from various domains), (c) a **fine-tuned model** for a specific domain (fine-tuned based on the one-for-all model) and (d) a **cross-domain model** pretrained only on LargeST (Liu et al., 2024b) (comprising 100.1 million collected samples). Cross-domain model will be evaluated on time series from other domains to verify the domain-transfer zero-shot capability in Section 4.3. Moreover, we evaluated BERT (Devlin et al., 2019) and LLaMA2 (Touvron et al., 2023) as backbones for NuwaTS (details in Appendix D.5).

**Baselines.** Following the configuration outlined in (Du et al., 2023), we evaluated two naive imputation methods: **Median**, where missing values are filled with the median value, and **Last**, where missing values are filled with the last previous observations. To ensure a fair comparison, all methods followed a unified pipeline[1]. We assessed two classic deep learning-based imputation models, **BRITS**(Cao et al., 2018) and **SAITS**(Du et al., 2023). We also compared our model with other foundation models for time series such as **DLinear**(Zeng et al., 2023), **PatchTST**(Nie et al., 2023), **iTransformer**(Liu et al., 2024c), **TimesNet**(Wu et al., 2022), **Autoformer**(Wu et al., 2021), **Fedformer**(Zhou et al., 2022), and **GPT4TS**(Zhou et al., 2024). We trained GPT4TS separately on single datasets with the number of layers set to 6 to align with NuwaTS. We trained all these models using a domain-specific approach. Since PatchTST is a transformer-based model with bi-directional attention, we also trained and tested it on the general fused dataset.

**Setups.** **We partitioned the dataset along the sensor (variable) dimension into training, validation, and test sets in a 1:1:1 ratio.** This division simulates the process of collecting relatively complete time series data from a few sensors or variables with lower missing rates, training a model with this data, and then using it to impute data from other sensors or variables that have higher missing rates. The input time series length $L$ is 96, and we trained all baselines under random missing rates (sampled from 0.1 to 0.9) and tested them separately under 9 missing rates: 0.1, 0.2,..., 0.9. We used a total of ten datasets for domain-specific training and mixed them into a general fused dataset for one-for-all training. Appendix A.1 shows the details of datasets. For the cross-domain model, we chose LargeST[2], a large-scale traffic dataset including a total of 8,600 time series and over 100 million samples, to train NuwaTS and conduct zero-shot experiments on other domains such as ETTs, weather and electricity.

### 4.1 MAIN RESULTS

We present the average MAE and MSE across different missing rates for specific, one-for-all, and fine-tuned NuwaTS models, along with other baselines, across 10 different datasets in Table 1 and Table 2, detailed results are shown in Appendix E. Furthermore, we visualize the average MSE across all datasets at various missing rates for the NuwaTS models and several baselines, as shown in Figure 5. On nearly all datasets, the NuwaTS model outperforms other domain-specific models. Moreover, we observed that the generalization ability of both NuwaTS and PatchTST (one-for-all) was further enhanced after training on a fused dataset spanning multiple domains, supporting the

---

[1]https://github.com/thuml/Time-Series-Library
[2]https://github.com/liuxu77/LargeST

Table 1: Imputation results are presented for ETTs, ECL, and weather datasets with nine test missing rates: 0.1, 0.2, ..., 0.9. All results are averaged across the nine different missing rates. A lower MSE or MAE signifies better imputation performance. **Green** highlights the best results, while **Yellow** indicates the second-best.

| Model | ETTh1 | | ETTh2 | | ETTm1 | | ETTm2 | | ECL | | Weather | |
|---|---|---|---|---|---|---|---|---|---|---|---|---|
| | MSE | MAE | MSE | MAE | MSE | MAE | MSE | MAE | MSE | MAE | MSE | MAE |
| Median | 0.723 | 0.611 | 0.728 | 0.472 | 0.699 | 0.584 | 0.744 | 0.464 | 1.010 | 0.834 | 0.998 | 0.497 |
| Last | 0.432 | 0.476 | 0.089 | 0.149 | 0.336 | 0.413 | 0.059 | 0.122 | 0.965 | 0.826 | 0.731 | 0.349 |
| Autoformer(2021) | 0.552 | 0.547 | 0.472 | 0.420 | 0.346 | 0.409 | 0.211 | 0.312 | 0.137 | 0.262 | 0.610 | 0.450 |
| Fedformer(2022) | 0.354 | 0.436 | 0.538 | 0.429 | 0.076 | 0.185 | 0.055 | 0.156 | 0.129 | 0.258 | 0.237 | 0.211 |
| DLinear(2023) | 0.356 | 0.414 | 0.245 | 0.301 | 0.274 | 0.357 | 0.266 | 0.351 | 0.317 | 0.419 | 0.355 | 0.309 |
| iTransformer(2024) | 0.639 | 0.572 | 0.369 | 0.376 | 0.352 | 0.413 | 0.356 | 0.416 | 0.092 | 0.200 | 0.515 | 0.412 |
| BRITS(2018) | 0.213 | 0.313 | 0.117 | 0.172 | 0.096 | 0.183 | 0.071 | 0.123 | 0.317 | 0.427 | 0.793 | 0.474 |
| TimesNet(2022) | 0.166 | 0.280 | 0.021 | 0.091 | 0.066 | 0.155 | 0.011 | 0.063 | 0.362 | 0.450 | 0.681 | 0.280 |
| PatchTST(2023) | 0.185 | 0.298 | 0.022 | 0.093 | 0.080 | 0.183 | 0.011 | 0.066 | 0.136 | 0.266 | 0.271 | 0.122 |
| SAITS(2023)[1] | 0.196 | 0.289 | 0.215 | 0.190 | 0.087 | 0.163 | 0.135 | 0.139 | 0.445 | 0.512 | 0.932 | 0.484 |
| GPT4TS(2024) | 0.196 | 0.290 | 0.025 | 0.092 | 0.078 | 0.161 | 0.012 | 0.063 | 0.296 | 0.401 | 0.939 | 0.321 |
| NuwaTS(specific) | 0.182 | 0.293 | 0.020 | 0.091 | 0.070 | 0.164 | 0.011 | 0.064 | 0.086 | 0.191 | 0.307 | 0.127 |
| PatchTST(one-for-all) | 0.178 | 0.288 | 0.019 | 0.088 | 0.075 | 0.172 | 0.011 | 0.065 | 0.121 | 0.243 | 0.230 | 0.116 |
| NuwaTS(one-for-all) | **0.164** | **0.263** | **0.018** | **0.084** | **0.064** | **0.147** | **0.010** | **0.060** | **0.085** | **0.186** | **0.206** | **0.088** |
| NuwaTS(fine-tuned) | **0.156** | **0.255** | **0.017** | **0.082** | **0.060** | **0.142** | **0.010** | **0.058** | **0.081** | **0.183** | **0.207** | **0.086** |

[1] We replace the MAE loss function in SAITS (Du et al., 2023) and BRITS (Cao et al., 2018) with MSE loss function for fair comparison.

Table 2: Imputation results on four PEMS datasets with the same setting as Table 1.

| Model | PEMS03 | | PEMS04 | | PEMS07 | | PEMS08 | |
|---|---|---|---|---|---|---|---|---|
| | MSE | MAE | MSE | MAE | MSE | MAE | MSE | MAE |
| Median | 0.691 | 0.609 | 0.742 | 0.645 | 0.755 | 0.646 | 0.734 | 0.653 |
| Last | 0.474 | 0.506 | 0.487 | 0.517 | 0.507 | 0.517 | 0.446 | 0.495 |
| Autoformer(2021) | 0.792 | 0.658 | 0.404 | 0.495 | 0.406 | 0.492 | 1.068 | 0.750 |
| Fedformer(2022) | 0.263 | 0.378 | 0.350 | 0.444 | 0.318 | 0.423 | 0.274 | 0.374 |
| DLinear(2023) | 0.262 | 0.390 | 0.266 | 0.392 | 0.259 | 0.387 | 0.268 | 0.393 |
| iTransformer(2024) | 0.108 | 0.237 | 0.131 | 0.259 | 0.099 | 0.225 | 0.162 | 0.291 |
| BRITS(2018) | 0.143 | 0.267 | 0.259 | 0.370 | 0.228 | 0.353 | 0.225 | 0.344 |
| TimesNet(2022) | 0.102 | 0.221 | 0.152 | 0.268 | 0.132 | 0.252 | 0.114 | 0.231 |
| PatchTST(2023) | 0.059 | 0.170 | 0.074 | 0.190 | 0.052 | 0.159 | 0.067 | 0.180 |
| SAITS(2023) | 0.157 | 0.286 | 0.271 | 0.380 | 0.228 | 0.349 | 0.250 | 0.364 |
| GPT4TS(2024) | 0.101 | 0.220 | 0.158 | 0.271 | 0.131 | 0.250 | 0.110 | 0.227 |
| NuwaTS(specific) | 0.049 | 0.149 | 0.058 | 0.159 | 0.040 | 0.129 | 0.057 | 0.156 |
| PatchTST(one-for-all) | 0.056 | 0.167 | 0.066 | 0.179 | 0.050 | 0.157 | 0.060 | 0.167 |
| NuwaTS(one-for-all) | **0.047** | **0.146** | **0.058** | **0.159** | **0.040** | **0.129** | **0.052** | **0.146** |
| NuwaTS(fine-tuned) | **0.047** | **0.146** | **0.058** | **0.159** | **0.040** | **0.129** | **0.052** | **0.146** |

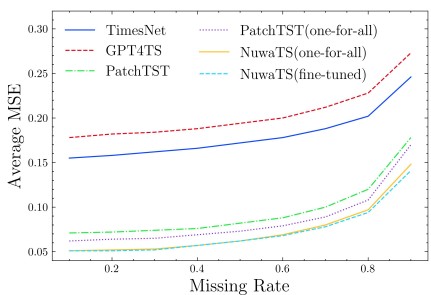

Figure 5: Main results across ten datasets under different missing rates.

presence of a scaling law (Kaplan et al., 2020) in time series imputation tasks. We visualized the imputation results in Appendix C.1. We also conducted experiments to verify NuwaTS on real-world dataset in Appendix D.1.

## 4.2 Imputation Results under Continuous Missing

In Section 4.1, we conduct a comprehensive evaluation of the performance of different approaches under purely random missing data scenarios. In real-world scenarios, missing data often occurs in a continuous manner (Du et al., 2024). Therefore, we also evaluated NuwaTS's performance under conditions of continuous missing data. The experimental setup is set as the following: Assuming a missing rate of $r$, we randomly selected a continuous segment of the time series with a length of $L \times r$ for masking. We evaluate the models trained on randomly missing data by directly testing them on datasets with continuous missing patterns.

Table 3: Imputation results on continuous missing data. The final results are averaged across 9 missing rates: 0.1, 0.2... 0.9. A lower MSE or MAE indicates better imputation performance. **Green**: the best.

| Model | ETTh1 | | ETTh2 | |
|---|---|---|---|---|
| | MSE | MAE | MSE | MAE |
| BRITS | 0.717 | 0.619 | 0.580 | 0.519 |
| SAITS | 0.519 | 0.469 | 0.318 | 0.250 |
| TimesNet | 0.497 | 0.494 | 0.110 | 0.154 |
| GPT4TS | 0.483 | 0.480 | 0.104 | 0.154 |
| NuwaTS | **0.465** | **0.455** | **0.095** | **0.141** |

As it is shown in Table 3, our findings reveal that NuwaTS exhibits strong domain adaptation capabilities and is highly robust in handling diverse missing patterns, further highlighting its generalizability across varying data missing patterns.

## 4.3 DOMAIN-TRANSFER ZERO-SHOT CAPABILITY

We validated the zero-shot capability of our model by directly evaluating a trained model on target data without further training. TimesNet and GPT4TS, which use a channel-dependent approach, can only perform zero-shot across datasets with the same number of variables, specifically the four ETT datasets in our study. In contrast, NuwaTS and PatchTST, as channel-independent approaches, support inference on datasets with different variable dimensions. We trained them on LargeST and then evaluated them on ECL and Weather datasets. The results in Table 4 shows NuwaTS exhibits superior zero-shot capability, achieving better performance on all cases. The reason may lie in the use of specialized missing data embedding and statistical embedding. Additionally, the language knowledge embedded in the PLM has strong generalization capabilities, whereas the PatchTST model trained on LargeST does not possess such strong capabilities.

Table 4: Zero shot evaluation results. All methods were trained on one dataset and zero shot to the other. A lower MSE or MAE indicates better imputation performance. Green : the best. /: model failed to work.

| METRICS | NUWATS | | PATCHTST | | GPT4TS | | TIMESNET | |
|---|---|---|---|---|---|---|---|---|
| | MSE | MAE | MSE | MAE | MSE | MAE | MSE | MAE |
| $ETTh1 \Rightarrow ETTh2$ | **0.023** | **0.091** | 0.024 | 0.092 | 0.029 | 0.095 | 0.026 | 0.095 |
| $ETTh1 \Rightarrow ETTm2$ | **0.013** | **0.072** | 0.013 | 0.073 | 0.015 | 0.073 | 0.014 | 0.072 |
| $ETTm1 \Rightarrow ETTh2$ | **0.021** | **0.094** | 0.022 | 0.096 | 0.027 | 0.093 | 0.024 | 0.096 |
| $ETTm1 \Rightarrow ETTm2$ | **0.011** | **0.063** | 0.011 | 0.067 | 0.013 | 0.064 | 0.013 | 0.066 |
| $LargeST \Rightarrow ECL$ | **0.338** | **0.366** | 0.385 | 0.433 | / | / | / | / |
| $LargeST \Rightarrow Weather$ | **0.217** | **0.087** | 0.236 | 0.121 | / | / | / | / |

## 4.4 FEW-SHOT DOMAIN-SPECIFIC FINE-TUNING

The ability to generalize to the target domain with a small amount of data is an important criterion for evaluating the generality of a model. We use $10\%$ and $1\%$ of the data for fine-tuning the cross-domain NuwaTS model which is trained on LargeST. The results shown in Table 5 indicate that domain-specific fine-tuning is particularly effective for fields with limited data. On the ETT datasets, we achieved the same results using only 10% of the data as we did using 100% of the data.

Table 5: Fine-tuned results from cross-domain NuwaTS model pre-trained on LargeST (Liu et al., 2024b). A lower MSE or MAE indicates better imputation performance. Green best, Yellow second best, and ↓ indicates fine-tuning effectiveness.

| METHODS | ECL | | ETTH1 | | ETTH2 | | ETTM1 | | ETTM2 | |
|---|---|---|---|---|---|---|---|---|---|---|
| | MSE | MAE | MSE | MAE | MSE | MAE | MSE | MAE | MSE | MAE |
| ZERO-SHOT | 0.338 | 0.366 | 0.212 | 0.306 | 0.021 | 0.089 | 0.066 | 0.145 | 0.011 | 0.059 |
| FINE-TUNING WITH 1% DATA | **0.284**↓ | **0.338**↓ | **0.205**↓ | **0.305**↓ | **0.021**↓ | 0.090 | **0.063**↓ | 0.147 | **0.010**↓ | 0.059 |
| FINE-TUNING WITH 10% DATA | **0.190**↓ | **0.292**↓ | **0.180**↓ | **0.282**↓ | **0.019**↓ | **0.086**↓ | **0.060**↓ | **0.142**↓ | **0.010**↓ | **0.058**↓ |
| FINE-TUNING WITH 100% DATA | **0.143**↓ | **0.253**↓ | **0.180**↓ | **0.280**↓ | **0.019**↓ | **0.085**↓ | **0.061**↓ | **0.143**↓ | **0.010**↓ | **0.058**↓ |

## 4.5 ABLATION STUDY

We conducted ablation experiments via zero-shot evaluation. We trained the default model, several ablated models on ETTh1 with 34650 samples and LargeST with more than 100.1 million samples. We then performed zero-shot evaluation on other datasets for out-of-domain evaluation. As shown in Table 6, after ablation of the Statistic Embedding, Missing Embedding, and contrastive learning module, the model's zero-shot ability decreases significantly, proving the necessity of these key components. Freezing the PLM backbone leads to poor performance. Finally, we discovered that when training the model from scratch without using weights pre-trained on NLP tasks, its zero-shot generalization ability on other datasets deteriorates significantly. This indicates that training the model on NLP tasks benefits its performance on time series tasks, demonstrating that cross-modality training is meaningful (Zhang et al., 2024).

Table 6: Ablation results of the model training on ETTh1 and LargeST and zero-shot on other data domain. A lower MSE or MAE indicates better imputation performance. **Green** : the best.

| Model | ETTH1 ⇒ ETTH2 | | ETTH1 ⇒ Weather | | LargeST ⇒ ECL | | LargeST ⇒ Weather | |
|---|---|---|---|---|---|---|---|---|
| | MSE | MAE | MSE | MAE | MSE | MAE | MSE | MAE |
| Default | 0.023 | 0.091 | 0.284 | 0.164 | 0.338 | 0.366 | 0.217 | 0.087 |
| w/o-StatisticEmbedding | 0.024 | 0.091 | 0.303 | 0.176 | 0.371 | 0.383 | 0.224 | 0.093 |
| w/o-ContrastiveLearning | 0.025 | 0.097 | 0.311 | 0.184 | 0.348 | 0.366 | 0.218 | 0.087 |
| w/o-MissingEmbedding | 0.028 | 0.101 | 0.331 | 0.193 | 0.338 | 0.368 | 0.221 | 0.092 |
| w/o-GPT2Weight[1] | 0.024 | 0.093 | 0.311 | 0.177 | 0.339 | 0.368 | 0.218 | 0.089 |
| FrozenBackbone | 0.027 | 0.095 | 0.344 | 0.207 | 0.533 | 0.560 | 0.290 | 0.131 |

[1] We trained NuwaTS from scratch without loading the pre-trained language weight.

When comparing the ablation results on ETTh1 and LargeST, where the former is a very small dataset and the latter is significantly larger. Our findings show that the specially designed tokens and contrastive learning modules provide more noticeable improvements on the smaller ETTh1 dataset. This suggests that for foundation models, data quantity might be more critical than the inductive biases introduced by specialized modules.

## 4.6 Enhancing Forecasting on Incomplete Time Series

Table 7: We trained and evaluated TimesNet (Wu et al., 2022) on incomplete time series with original missing rates of 0.2 and 0.5, comparing its forecasting performance on data imputed by PatchTST and NuwaTS. We set the length of input sequence and forecasting both to 96. A lower MSE or MAE indicates better forecasting performance.

| Methods | ETTH1 | | ETTH2 | | ETTM1 | | ETTM2 | |
|---|---|---|---|---|---|---|---|---|
| | MSE | MAE | MSE | MAE | MSE | MAE | MSE | MAE |
| Complete Data | 0.384 | 0.402 | 0.340 | 0.374 | 0.338 | 0.375 | 0.187 | 0.267 |
| **Missing Rate: 0.2** | | | | | | | | |
| Default | 0.457 | 0.450 | 0.353 | 0.389 | 0.389 | 0.404 | 0.200 | 0.284 |
| +PatchTST | 0.439 | 0.445 | 0.346 | 0.382 | 0.353 | 0.388 | 0.189 | 0.270 |
| +NuwaTS | 0.428 | 0.437 | 0.341 | 0.378 | 0.344 | 0.381 | 0.190 | 0.269 |
| **Missing Rate: 0.5** | | | | | | | | |
| Default | 0.585 | 0.515 | 0.370 | 0.403 | 0.572 | 0.496 | 0.238 | 0.318 |
| +PatchTST | 0.450 | 0.452 | 0.354 | 0.389 | 0.357 | 0.391 | 0.189 | 0.270 |
| +NuwaTS | 0.441 | 0.446 | 0.344 | 0.382 | 0.345 | 0.383 | 0.190 | 0.270 |

In practical applications, forecasting models often have to train on incomplete data, which can significantly impair their performance. To address this issue, we applied pre-trained imputation models to impute the incomplete data before training the forecasting model. We chose TimesNet (Wu et al., 2022) as the forecasting model and used four ETT datasets for training. We randomly discarded 20% and 50% of the data in the training sets of four ETT datasets.

To prevent data leakage during imputation, we used cross-domain NuwaTS and PatchTST pre-trained on LargeST to impute the training data. Finally, we trained TimesNet (Wu et al., 2022) on the imputed time series and tested it on the complete time series.

As shown in Table 7, the data imputed by NuwaTS improved the forecasting model's performance across most metrics. Additionally, the imputed data from NuwaTS was of higher quality for down-stream tasks compared to PatchTST, demonstrating that NuwaTS can effectively address the issue of incomplete time series data in practical applications.

## 4.7 Fine-tuning Imputation Model to Forecasting Model

By directly inserting the masked padding tokens $\mathbf{p}$ where $\mathbf{p} \in \mathbb{R}^{M \times D}$ after the original input embeddings $\mathbf{E}_i \in \mathbb{R}^{(N+2) \times D}$, we get $\mathbf{E}_i^f = [\mathbf{k}, \mathbf{z}_{i,(v_g)}, \mathbf{E}_{i,(p)}, \mathbf{p}]$ where $\mathbf{E}_i^f \in \mathbb{R}^{(N+2+M) \times D}$. The model automatically imputes the future $M$ patches, effectively transforming into a forecasting model. We discard the prefixal part and obtain the final forecasting represenation $\mathbf{h}_i^{(Layer)} \in \mathbb{R}^{M \times D}$. Then $\mathbf{h}_i^{(Layer)}$ pass through the output layer and we get the final forecasting results. In order to further help NuwaTS adapt to the forecasting task, we conducted two types of fine-tuning module based on the cross-domain model, the first one using two-layer MLP as the domain-transfer layer which

Table 8: Forecasting results. We set the length of input sequence and forecasting both to 96. Forecasting results from other baselines come from (Wu et al., 2022; Liu et al., 2024c; Cai et al., 2024). We fine-tuned the cross-domain model in order to avoid data leakage. A lower MSE or MAE indicates a better performance. **Green** : the best, Yellow : the second best.

| MODEL | ETTH1 MSE | ETTH1 MAE | ETTH2 MSE | ETTH2 MAE | ETTM1 MSE | ETTM1 MAE | ETTM2 MSE | ETTM2 MAE | ECL MSE | ECL MAE | WEATHER MSE | WEATHER MAE |
|---|---|---|---|---|---|---|---|---|---|---|---|---|
| AUTOFORMER(2021) | 0.449 | 0.459 | 0.346 | 0.388 | 0.505 | 0.475 | 0.255 | 0.339 | 0.201 | 0.317 | 0.266 | 0.336 |
| FEDFORMER(2022) | 0.395 | 0.424 | 0.358 | 0.397 | 0.379 | 0.419 | 0.203 | 0.287 | 0.193 | 0.308 | 0.217 | 0.296 |
| TIMESNET(2022) | 0.384 | **0.402** | 0.340 | 0.374 | 0.338 | 0.375 | 0.187 | 0.267 | 0.168 | 0.272 | 0.172 | 0.220 |
| DLINEAR(2023) | 0.397 | 0.412 | 0.333 | 0.387 | 0.345 | 0.372 | 0.193 | 0.292 | 0.197 | 0.282 | 0.196 | 0.255 |
| PATCHTST(2023) | 0.460 | 0.447 | 0.308 | 0.355 | 0.352 | 0.374 | 0.183 | 0.270 | 0.190 | 0.296 | 0.186 | 0.227 |
| MSGNET(2024) | 0.390 | 0.411 | 0.328 | 0.371 | **0.319** | 0.366 | **0.177** | **0.262** | 0.165 | 0.274 | **0.163** | **0.212** |
| ITRANSFORMER(2024) | 0.386 | 0.405 | 0.297 | 0.349 | 0.334 | 0.368 | 0.180 | 0.264 | 0.148 | 0.240 | 0.174 | 0.214 |
| NUWATS(**WITHOUT** INTER-VARIABLE CORRELATION) | 0.375 | 0.404 | 0.292 | 0.348 | 0.319 | 0.360 | 0.185 | 0.268 | 0.183 | 0.267 | 0.172 | 0.215 |
| NUWATS(**WITH** INTER-VARIABLE CORRELATION) | 0.374 | 0.403 | 0.308 | 0.357 | 0.314 | 0.357 | 0.184 | 0.267 | 0.151 | 0.242 | 0.179 | 0.221 |

accounts for only 9.35% of the model's total parameters (details in Appendix A.3), and the other one computing the inter-variable correlation. (details in Appendix B.4) We visualized the forecasting results in Appendix C.2.

## 5  CONCLUSION

In this paper, we address the challenging problem of cross-variable and cross-domain generalization in time series imputation. We introduce NuwaTS, a time series imputation model built on a PLM backbone that explicitly captures patch-wise statistical information and missing patterns. Additionally, we propose an efficient domain-specific fine-tuning technique that allows the model to seamlessly adapt from an all-in-one imputation model to a domain-specific one with minimal cost, and even transform into a forecasting model when needed. To benchmark the model's performance, we design a novel variable-wise train/validation/test partitioning strategy, providing a rigorous evaluation framework. Experimental results show that our approach significantly outperforms state-of-the-art methods, demonstrating strong adaptability across diverse domains and missing rates, even in zero-shot scenarios. To the best of our knowledge, this is the first foundational time series imputation model capable of generalizing across domains. We believe our method sets a new benchmark in the field and provides valuable insights for future research.

## 6  REPRODUCIBILITY STATEMENT

This paper is reproducible. Experimental details about all empirical results described in this paper are provided in Appendix. Additionally, we provide the PyTorch code for reproducing our results at https://anonymous.4open.science/r/NuwaTS-85FB. The dataset used in this paper is available at https://github.com/thuml/Time-Series-Library and https://github.com/liuxu77/LargeST. Formal proofs under a rigorous setting of all our theoretical results are provided in Appendix A.2 and B.

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

# A EXPERIMENTAL DETAILS

## A.1 DATASET DETAILS

Our research encompasses experimental evaluations utilizing a selection of ten popular multivariate datasets, including (1) **ETT**[3] reports seven factors of electricity transformers, encompassing four subsets (ETTm1, ETTm2, ETTh1, ETTh2). These are divided into two categories based on temporal granularity, with data recorded at intervals of 15 minutes (m) and 1 hour (h), respectively. (2) **ECL**[4] records the hourly electricity consumption of 321 customers. (3) **Weather**[5] comprises 21 meteorological factors, recorded at ten-minute intervals throughout the year 2020, including variables such as temperature and humidity. (4) **PEMS** encompasses four datasets (03, 04, 07, and 08), each of which collects data on California's public traffic network at a frequency of every five minutes. For the LargeST (Liu et al., 2024b), which consists of 8,600 sensors and spans from 2017 to 2021, we only selected the 2019 data for pre-training NuwaTS in this paper.

For the eleven datasets mentioned above, we split the data from the variable dimension rather than the time dimension, using a 1:1:1 ratio for training, validation, and test datasets. This approach is necessary because some baselines use a channel-dependent method, which requires fixed input dimensions for time series.

For the general fused datasets, we processed each time variable separately. We sliced time variables from all datasets (ETT, Weather, ECL, PEMS) of varying lengths with a stride of 1, creating a combined dataset with 17.6 million time segments of fixed length. In this study, we set it to 96. These segments were then randomly masked and used for model training.

Using the same approach, we sliced LargeST into 100.1 million segments to train NuwaTS and PatchTST. This was done for zero-shot evaluation and fine-tuning for forecasting tasks on ETT, weather, and ECL, ensuring no data leakage.

## A.2 IMPLEMENTATION DETAILS

Regarding the training details of our model compared to others, we mainly followed the imputation task configuration, including optimizer, learning rate, and early stop strategy in the https://github.com/thuml/Time-Series-Library for fair comparisons. We trained three different PLM backbones: GPT2 (Radford et al., 2019), BERT (Devlin et al., 2019), and LLaMA (Touvron et al., 2023). All the comparison models, as well as GPT2 and BERT, were trained on NVIDIA RTX 3090-24G GPUs. LLaMA-version (the first six layers) was trained on an NVIDIA A6000-48G GPU.

## A.3 DIFFERENT PARAMETER-EFFICIENT TRAINING STRATEGY

We trained NuwaTS using three different PLM backbones. Table 9 presents the detailed training specifics for each experiment and backbone, including fine-tuned network layers, memory usage, and the number of parameters.

---

[3] https://github.com/zhouhaoyi/ETDataset
[4] https://archive.ics.uci.edu/ml/datasets/ElectricityLoadDiagrams20112014
[5] https://www.bgc-jena.mpg.de/wetter/

Table 9: Different parameter-efficient training strategies on GPT2, BERT, and LLaMA2.

| Task | Input Size | Backbone (number of layers) | Training Module | Memory Occupation (GB) | Training / Total Parameter (M) |
|------|-----------|----------------------------|-----------------|------------------------|-------------------------------|
| Imputation | (512,96,1) | GPT2 (3) | LN, WPE, IN&OUT, FFN | 2.7 | 20.3 / 66.0 |
| Imputation | (512,96,1) | GPT2 (6) | LN, WPE, IN&OUT, FFN | 4.6 | 38.0 / 90.8 |
| Imputation | (512,96,1) | GPT2 (9) | LN, WPE, IN&OUT, FFN | 6.4 | 55.8 / 115.6 |
| Imputation | (512,96,1) | GPT2 (12) | LN, WPE, IN&OUT, FFN | 8.3 | 73.5 / 140.4 |
| Imputation | (512,96,1) | LLaMA2 (6) | LN, IN&OUT | 19.5 | 6.3 / 1351.7 |
| Imputation | (128,96,1) | LLaMA2 (6) | LN, IN&OUT, FFN | 20.3 | 818.0 / 1351.7 |
| Imputation | (512,96,1) | BERT (6) | LN, WPE, IN&OUT, FFN | 3.1 | 34.1 / 68.2 |
| Imputation (fine-tuned) | (512,96,1) | GPT2 (6) | DTL | 2.4 | 7.7 / 90.8 |
| Forecasting (w/o-OutputLayer) | (512,96,1) | GPT2 (6) | DTL, LN, WPE | 3.8 | 8.5 / 90.8 |
| Forecasting | (512,96,1) | GPT2 (6) | DTL, LN, WPE, OUT | 3.8 | 8.9 / 90.8 |

LN: LayerNorm Layer, WPE: Word Position Encoding, IN&OUT: EmbedLayer&OutputLayer
FFN: Feed-Forward Network, DTL: Domain-Transfer Layer&prefix

# B  TRAINING ALGORITHM AND MATHEMATICAL FORMULA

## B.1  ALGORITHM

---

**Algorithm 1** One-for-all Model Training

---

1: Given the relatively complete time series datasets $\mathcal{D} = \{\mathbf{x}^1, \mathbf{x}^2, \ldots, \mathbf{x}^N\}$ from diverse set of domains.
2: **for** $\mathbf{x}_i \in \mathbb{R}^L \leftarrow \mathcal{D}$ **do**
3:     Generate two random mask $\mathbf{m}_{i,1} \in \mathbb{R}^L, \mathbf{m}_{i,2} \in \mathbb{R}^L$
4:     $\hat{\mathbf{x}}_{i,1}, \hat{\mathbf{x}}_{i,2} \leftarrow \mathbf{x}_i \times \mathbf{m}_{i,1}, \mathbf{x}_i \times \mathbf{m}_{i,2}$
5:     $\mathbf{Z}_{i,(p)} \leftarrow \text{Embed}(\hat{\mathbf{x}}_i)$
6:     $\mathbf{E}_{i,(p)} \leftarrow \mathbf{Z}_{i,(p)} + \mathbf{Z}_{i,(v_p)} + \mathbf{z}_{i,(m)} \times \mathbf{r}_i$
7:     $\mathbf{E}_i \leftarrow [\mathbf{k}, \mathbf{z}_{i,(v_g)}, \mathbf{E}_{i,(p)}]$
8:     Obtain the last hidden states $\mathbf{h}_i^{(\text{Layer})} \leftarrow \text{PLM}(\mathbf{E}_i)$
9:     Obatin the imputed series $\mathbf{o}_i \leftarrow \text{OutputLayer}\left(\mathbf{h}_i^{(\text{Layer})}\right)$
10:     Update $\boldsymbol{\Phi}$ by gradients for $\mathcal{L}_{mse}\left(\mathbf{o}_{i,1}, \mathbf{x}_i\right) + \mathcal{L}_{mse}\left(\mathbf{o}_{i,2}, \mathbf{x}_i\right) + \alpha \mathcal{L}_{cl}\left(\mathbf{h}_{i,1}^{(\text{Layer})}, \mathbf{h}_{i,2}^{(\text{Layer})}\right)$
11: **end for**

---

**Algorithm 2** Domain Specific Fine-tuning

---

1: Initialize continuous prefix $\mathcal{P} \in \mathbb{R}^{2 \times \text{Layer} \times d}$
2: Obtain $\hat{\mathcal{K}} \in \mathbb{R}^{2 \times \text{Layer} \times d} \leftarrow \text{DomainTrans}(\mathbf{k})$
3: WITH NO GRAD:
4: **for** $n \leftarrow 0$ to Layer **do**
5:     Obtain hidden state $\mathbf{h}^{(n-1)}$ from $\text{PLMLayer}^{(n-1)}$
6:     Obtain $[\mathbf{Key}_p, \mathbf{Value}_p] \leftarrow \mathcal{P}^{(n)} + \beta \hat{\mathcal{K}}^{(n)}$ where $\mathcal{P}^{(n)} \in \mathbb{R}^{2 \times d}$
7:     Obtain $\mathbf{Key} \leftarrow \text{Concat}\left(\mathbf{Key}_p; \mathbf{Key}^{(n)}\right)$
8:     Obtain $\mathbf{Value} \leftarrow \text{Concat}\left(\mathbf{Value}_p; \mathbf{Value}^{(n)}\right)$
9:     Obtain $\mathbf{h}^{(n)} \leftarrow \text{PLMLayer}^{(n)}\left(\mathbf{h}^{(n-1)}, \mathbf{Key}, \mathbf{Value}\right)$
10: **end for**
11: Obtain $\mathbf{h}^{(\text{Layer})}$

---

---

**Algorithm 3** Fine-tuning with inter-variable correlation

---

1: Initialize $\mathbf{X} \in \mathbb{R}^{N \times T}$, where $N$ is the number of variables and $T$ is the time steps.
2: **for** $n = 1$ to Layer **do**
3:   Map each variable $\mathbf{x}_i \in \mathbf{X}$ to a token: $\mathbf{z}_i^{(n)} \in \mathbb{R}^{d_{\text{light}}}$, resulting in a token sequence $\mathbf{Z}^{(n)} \in \mathbb{R}^{N \times d_{\text{light}}}$ for layer $n$
4:   Apply lightweight transformer at layer $n$ to capture inter-variable correlations:
5:   $\mathbf{h}^{(n)} \leftarrow \text{TransformerLayer}^{(n)}(\mathbf{Z}^{(n)})$, where $\mathbf{h}^{(n)} \in \mathbb{R}^{N \times d_{\text{light}}}$
6:   Pass the hidden state $\mathbf{h}^{(n)}$ through a linear layer:
7:   $\hat{\mathcal{K}}^{(n)}, \hat{\mathcal{V}}^{(n)} \leftarrow \text{Linear}(\mathbf{h}^{(n)})$, where $\hat{\mathcal{K}}^{(n)}, \hat{\mathcal{V}}^{(n)} \in \mathbb{R}^{N \times d_{\text{PLM}}}$
8:   $\mathcal{P}^{(n)} \leftarrow \text{Concat}(\hat{\mathcal{K}}^{(n)}, \hat{\mathcal{V}}^{(n)})$, where $\mathcal{P}^{(n)} \in \mathbb{R}^{N \times 2 \times d_{\text{PLM}}}$
9: **end for**
10: The generated prefix for all layers $\mathcal{P} \in \mathbb{R}^{\text{Layer} \times N \times 2 \times d_{\text{PLM}}}$ is then used for PLM fine-tuning.

---

## B.2 INSTANCE NORM

We applied reversible instance normalization (RevIN) (Kim et al., 2021) to individual series before splitting them into patches. Each series was normalized to have zero mean and unit standard deviation, with the original mean and standard deviation stored for later reversal, thereby reducing the domain distribution shift caused by non-stationary information. The details are as follows:

$$
\mathbb{E}_t \left[ \mathbf{x}_i \right] = \frac{1}{L} \sum_{j=1}^{L} \mathbf{x}_{i,j},
$$

$$
\text{Var} \left[ \mathbf{x}_i \right] = \frac{1}{L} \sum_{j=1}^{L} \left( \mathbf{x}_{i,j} - \mathbb{E}_t \left[ \mathbf{x}_i \right] \right)^2 , \qquad (2)
$$

$$
\hat{\mathbf{x}}_i = \left( \frac{\mathbf{x}_i - \mathbb{E}_t \left[ \mathbf{x}_i \right]}{\sqrt{\text{Var} \left[ \mathbf{x}_i \right] + \epsilon}} \right) .
$$

## B.3 CONTRASTIVE LEARNING LOSS

Given a time patch represenation as a query $q$, and a set of keys $\mathbb{K} = \{k_0, k_1, \ldots\}$, including a positive target $k_+$. We employ the InfoNCE (Oord et al., 2018) as an auxiliary loss function to optimize the model's performance:

$$
\mathcal{L}_{cl} = - \log \frac{\exp\left(q^T W k_+\right)}{\exp\left(q^T W k_+\right) + \sum_{i=0}^{K-1} \exp\left(q^T W k_i\right)}, \qquad (3)
$$

where we employ bi-linear inner product and $W$ is a learnable matrix.

## B.4 FINE-TUNING
WITH INTER-VARIABLE CORRELATION

Unlike the two-layer MLP used in the channel-independent paradigm in Section 3.3, inspired by TimeXer (Wang et al., 2024b) and iTransformer (Liu et al., 2024c), we map each time series variable into a token. We then apply a stack of lightweight transformer layers, with the same number

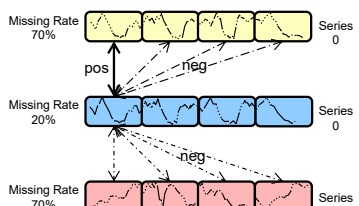

Figure 6: Illustration of contrastive learning: Representations of each patch under different masks are treated as positive samples for each other, while representations from other time series and different patches from the same sequence under varying masks are considered negative samples.

of layers as the PLM, to obtain the hidden states from each layer. These hidden states are passed through a linear layer to be directly mapped to $\hat{\mathcal{K}} \in \mathbb{R}^{N \times 2 \times \text{Layer} \times D}$, thus generating the prefix. This prefix now contains inter-variable correlation information, addressing the limitation of NuwaTS in modeling relationships between variables.

# C VISUALIZATION

## C.1 IMPUTATION VISUALIZATION

We visualized the imputation results of NuwaTS in Figure 7, Figure 8, Figure 9 and Figure 10.

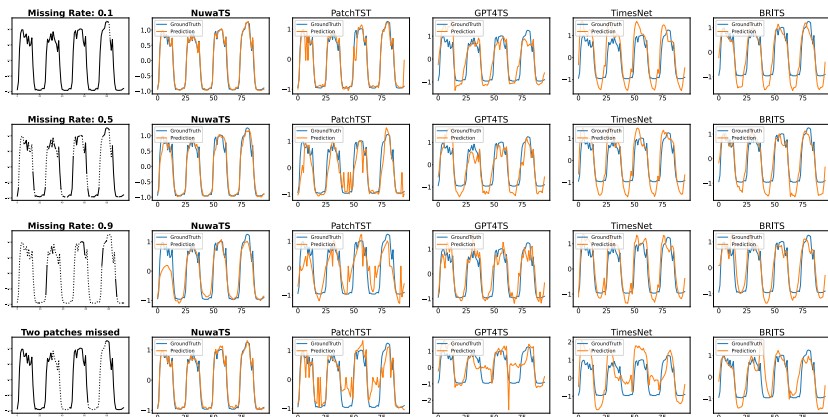

Figure 7: Case visualization when missing rate set to 0.1, 0.5, 0.9. We also visualized the case where the two patches are missing.

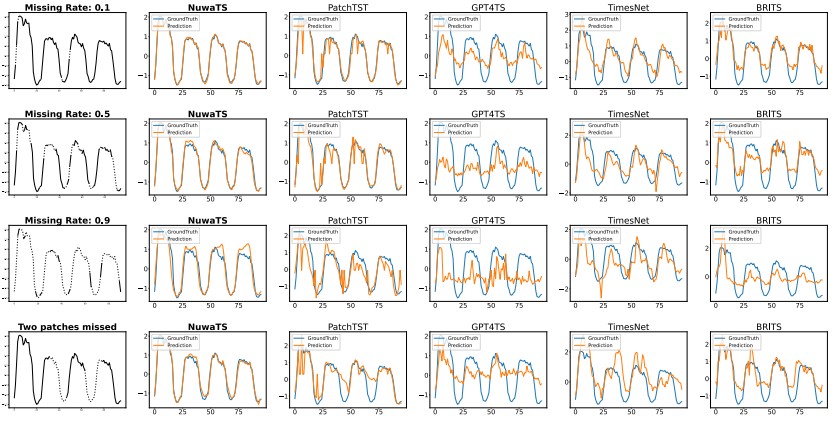

Figure 8: Case visualization when missing rate set to 0.1, 0.5, 0.9. We also visualized the case where the two patches are missing.

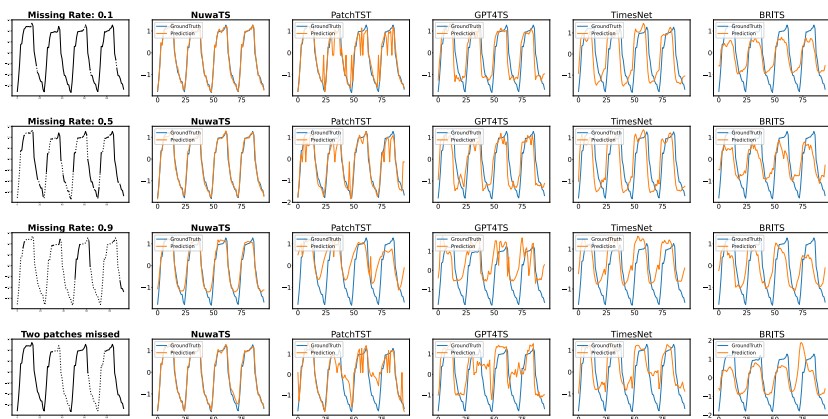

Figure 9: Case visualization when missing rate set to 0.1, 0.5, 0.9. We also visualized the case where the two patches are missing.

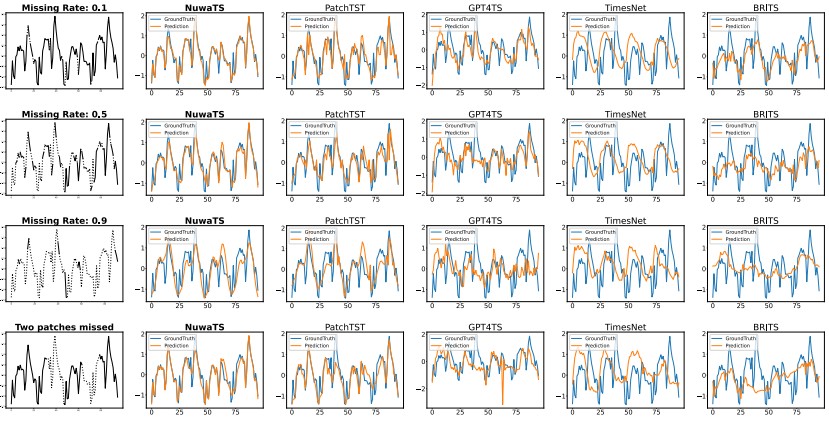

Figure 10: Case visualization when missing rate set to 0.1, 0.5, 0.9. We also visualized the case where the two patches are missing.

## C.2 FORECASTING VISUALIZATION

We visualized the forecasting results of fine-tuned NuwaTS whose backbone had not undergone any forecasting training in Figure 11.

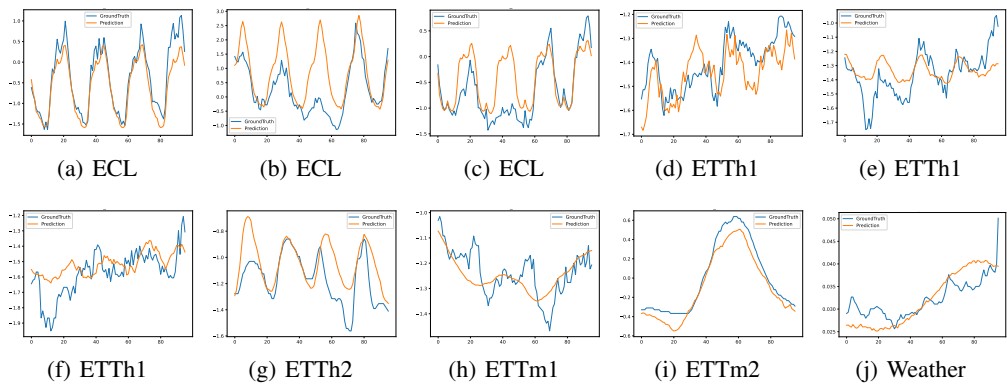

(a) ECL    (b) ECL    (c) ECL    (d) ETTh1    (e) ETTh1

(f) ETTh1    (g) ETTh2    (h) ETTm1    (i) ETTm2    (j) Weather

Figure 11: Case visualization when sequence length set to 96 and forecasting length set to 96.

### C.3 DOMAIN RECOGNITION

We also visualized the domain recognition ability of NuwaTS in Figure 12. By directly appending domain-specific embeddings $\mathbf{k}$ again to the end of the input embedding $\mathbf{E}_i \in \mathbb{R}^{(N+2) \times D}$, we get $\hat{\mathbf{E}}_i = [\mathbf{k}, \mathbf{z}_{i,(v_g)}, \mathbf{E}_{i,(p)}, \mathbf{k}]$, where $\hat{\mathbf{E}}_i \in \mathbb{R}^{D \times (N+3)}$. We feed the $\hat{\mathbf{E}}_i$ into the NuwaTS (one-for-all) model, and get the final representation $\hat{\mathbf{h}}_i^{(\text{Layer})} \in \mathbb{R}^{(N+3) \times D}$. Then We extract the last embeddings $\mathbf{k}' \in \mathbb{R}^D$ from $\hat{\mathbf{h}}_i^{(\text{Layer})}$ which is mixed with the patch embeddings in front of the input embeddings, thus exhibiting distinct domain characteristics. We collected $N$ time series from different domains and extracted $\mathbf{k}'$. After applying the t-SNE (Van der Maaten & Hinton, 2008) method to reduce the dimension to 2, we obtained scattered points $\mathbf{k}'^p \in \mathbb{R}^{N \times 2}$ and visualized them. The scattered points corresponding to time series from the same domain tend to cluster together, which indicates that NuwaTS can recognize domain information.

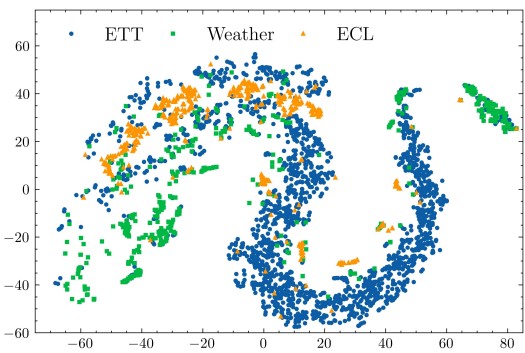

Figure 12: Illustration of domain recognition.

## D EXTRA EXPERIMENTS

### D.1 REAL-WORLD DATASET EVALUATION

We conducted experiments on Beijing Multi-Site Air-Quality (Zhang et al., 2017) in Table 10 following pipeline in SAITS (Du et al., 2023) where we observed that, despite not using inter-series correlations and the model not being directly trained on the data, the performance of the pre-trained one-for-all NuwaTS was close to that of SAITS, surpassing both BRITS and GP-VAE (Fortuin et al., 2020).

Table 10: Imputation results on the Air-Quality dataset with missing values. 10% of the observed values in the validation set and test set are held out as ground truth for evaluation. A lower RMSE or MAE indicates better imputation performance. Green : the best.

| MODEL | AIR-QUALITY | |
| --- | --- | --- |
| | RMSE | MAE |
| GP-VAE | 0.614 | 0.268 |
| BRITS | 0.525 | 0.153 |
| SAITS | 0.518 | **0.137** |
| NUWATS (ZERO-SHOT) | 0.370 | 0.151 |
| NUWATS (DOMAIN FINE-TUNED) | **0.363** | 0.144 |

## D.2 ABLATION STUDY OF DOMAIN-SPECIFIC FINE-TUNING

We also verified the design of domain-specific fine-tuning strategy of NuwaTS in Table 11. When the randomly initialized prefix key&value and the knowledge-transfer layer were ablated, the model's fine-tuning performance showed a significant decline which indicates that domain-transfer layer retains the learned knowledge while newly added prefix key&value provide flexibility in transferring to specific domain. Additionally, we find that merely appending the prefix to the input layer and fine-tuning it significantly reduces the effectiveness of fine-tuning. Inserting the prefix into each layer of the PLM enhances the prefix's representation capacity during domain transfer (Liu et al., 2022).

Table 11: Domain-specific fine-tuning ablation study on ETTs based on the model pre-trained on general fused dataset. A lower MSE or MAE indicates better imputation performance. Green : the best.

| MODEL | ETTH1 | | ETTH2 | | ETTM1 | | ETTM2 | |
| --- | --- | --- | --- | --- | --- | --- | --- | --- |
| | MSE | MAE | MSE | MAE | MSE | MAE | MSE | MAE |
| DOMAIN-SPECIFIC FINE-TUNING | **0.156** | **0.255** | **0.017** | **0.082** | **0.060** | **0.142** | **0.010** | **0.058** |
| W/O RANDOM INITIALIZED PREFIX KEY&VALUE | 0.160 | 0.260 | 0.019 | 0.085 | 0.063 | 0.149 | 0.010 | 0.060 |
| W/O DOMAIN-TRANSFER LAYER | 0.158 | 0.256 | 0.018 | 0.083 | 0.060 | 0.142 | 0.010 | 0.058 |
| ONLY APPLY TO INPUT EMBEDDING | 0.163 | 0.263 | 0.018 | 0.084 | 0.062 | 0.145 | 0.010 | 0.060 |
| NO FINE-TUNING | 0.164 | 0.263 | 0.018 | 0.084 | 0.064 | 0.147 | 0.010 | 0.060 |

## D.3 LEARNING FROM INCOMPLETE TRAINING DATA

Table 12: When model training on incomplete data with 0.2 and 0.5 original missing rate. The comparison between NuwaTS and PatchTST. A lower MSE or MAE indicates a better imputation performance.

| METHODS | ETTH1 | | ETTH2 | | ETTM1 | | ETTM2 | | ECL | | WEATHER | |
| --- | --- | --- | --- | --- | --- | --- | --- | --- | --- | --- | --- | --- |
| | MSE | MAE | MSE | MAE | MSE | MAE | MSE | MAE | MSE | MAE | MSE | MAE |
| DEFAULT | 0.164 | 0.263 | 0.018 | 0.084 | 0.064 | 0.147 | 0.010 | 0.060 | 0.085 | 0.186 | 0.206 | 0.088 |
| NUWATS-ORIGIN MISSING RATE 0.2 | 0.169 | 0.276 | 0.018 | 0.085 | 0.070 | 0.160 | 0.010 | 0.061 | 0.130 | 0.233 | 0.214 | 0.099 |
| NUWATS-ORIGIN MISSING RATE 0.5 | 0.170 | 0.274 | 0.020 | 0.086 | 0.077 | 0.167 | 0.010 | 0.062 | 0.160 | 0.258 | 0.228 | 0.114 |
| PATCHTST | 0.178 | 0.288 | 0.019 | 0.088 | 0.075 | 0.172 | 0.011 | 0.065 | 0.121 | 0.243 | 0.230 | 0.116 |
| PATCHTST-ORIGIN MISSING RATE 0.2 | 0.185 | 0.295 | 0.019 | 0.089 | 0.079 | 0.179 | 0.011 | 0.066 | 0.150 | 0.264 | 0.239 | 0.118 |
| PATCHTST-ORIGIN MISSING RATE 0.5 | 0.202 | 0.316 | 0.021 | 0.093 | 0.079 | 0.181 | 0.011 | 0.067 | 0.293 | 0.387 | 0.244 | 0.124 |

We simulated the training of the model on incomplete time series by randomly omitting 20% and 50% of the original training data and then tested its ability of resisting disturbance. The experimental results indicate that NuwaTS has strong resilience and can achieve good generalization performance when trained on data with a high missing rate (details shown in Table 12).

## D.4 THE EFFECT OF THE NUMBER OF THE GPT2 LAYER.

Through experiments, we tested the impact of the number of GPT-2 layers and patch size on the final results in Figure 13. The experimental results showed that using 6 GPT-2 layers yielded the best performance.

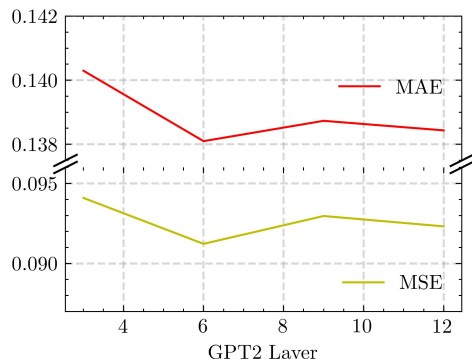

Figure 13: The effect of the number of GPT2 layers on Imputation Performance.

### D.5 EFFECT OF THE TYPE OF PLM BACKBONE

Table 13: Effect of different PLM backbone. A lower MSE or MAE indicates a better imputation performance. **Green** : the best.

| MODEL | ETTH1 MSE | ETTH1 MAE | ETTH2 MSE | ETTH2 MAE | ETTM1 MSE | ETTM1 MAE | ETTM2 MSE | ETTM2 MAE | ECL MSE | ECL MAE | WEATHER MSE | WEATHER MAE |
|---|---|---|---|---|---|---|---|---|---|---|---|---|
| GPT2 | **0.164** | **0.263** | **0.018** | 0.084 | **0.064** | **0.147** | **0.010** | 0.060 | **0.085** | **0.186** | **0.206** | **0.088** |
| BERT | 0.169 | 0.267 | 0.018 | 0.084 | 0.065 | 0.148 | 0.010 | **0.060** | 0.091 | 0.193 | 0.217 | 0.091 |
| LLAMA2 | 0.170 | 0.272 | 0.018 | **0.084** | 0.067 | 0.152 | 0.010 | 0.060 | 0.096 | 0.200 | 0.210 | 0.090 |

We experimented with three different backbones in Table 13: GPT2 (Radford et al., 2019), BERT (Devlin et al., 2019), and LLaMA (Touvron et al., 2023), uniformly using the first six layers of each. Among them, BERT is a transformer with bi-directional attention, while the other two, GPT2 and LLaMA, employ causal attention.

Using GPT-2 as the backbone outperformed the much larger LLaMA2, likely due to the relatively smaller amount of training data compared to LLaMA2's massive parameter count.

Additionally, LLaMA2's large embedding dimensions produced sparse features and the slow speed of LLaMA2 during training and evaluation makes deployment on edge devices challenging.

BERT, with the similar size as GPT-2, mainly differs in its bidirectional attention. It slightly lags behind GPT-2 due to the causal nature of time series data. However, BERT's bidirectional attention allows the domain-specific embedding to capture domain information from the following embed patches.

Based on the BERT model pre-trained on general fused dataset, We have $\mathbf{E}_i = [\mathbf{k}, \mathbf{z}_{i,(v_g)}, \mathbf{E}_{i,(p)}]$ before feeding into the backbone, where $\mathbf{E}_i \in \mathbb{R}^{D \times (N+2)}$. We feed the $\mathbf{E}_i$ into the backbone, and get the final representation $\mathbf{h}_i^{(\text{Layer})} \in \mathbb{R}^{(N+2) \times D}$. Then We extract the first embedding $\mathbf{k}' \in \mathbb{R}^D$ from $\mathbf{h}_i^{(\text{Layer})}$ and take it as [CLS] token (Devlin et al., 2019) which is mixed with the patch embeddings through bidirectional attention mechanism. With same approach as Appendix C.3, we obtain scattered points $\mathbf{k}'^p \in \mathbb{R}^{N \times 2}$ and visualize them in Figure 14.

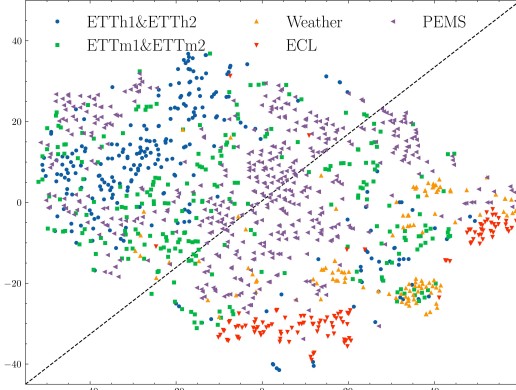

Figure 14: Domain information visualization. We take the model's first output embedding as the [CLS] token, which carries domain information from following sequences. We then use the t-SNE (Van der Maaten & Hinton, 2008) method to visualize the domain information from different datasets.

## D.6 EMBED WITH SIMPLE LINEAR LAYER VS. EMBED WITH TEXT-ALIGNMENT.

For tokenization, we employ a simple linear layer to embed patches. We acknowledge that recent research (Jin et al., 2024) has proposed utilizing Patch Reprogramming mechanism to align time patches with PLM's pre-trained word embeddings, thereby activating the model's time series understanding and reasoning capabilities. However, we contend that, due to the high proportion of missing values in the input masked patches and the varying locations of these missing values, modality alignment is not effective in representing such complex incomplete time series. Table 14 shows that simple linear embedding is better than the text-alignment strategy.

Table 14: Embedding methods comparison.

| MODEL | ETTH1 MSE | ETTH1 MAE | ETTH2 MSE | ETTH2 MAE | ETTM1 MSE | ETTM1 MAE | ETTM2 MSE | ETTM2 MAE | ECL MSE | ECL MAE | WEATHER MSE | WEATHER MAE |
|---|---|---|---|---|---|---|---|---|---|---|---|---|
| SIMPLE LINEAR LAYER | 0.164 | 0.263 | 0.018 | 0.084 | 0.064 | 0.147 | 0.010 | 0.060 | 0.085 | 0.186 | 0.206 | 0.088 |
| EMBED WITH TEXT-ALIGNMENT (JIN ET AL., 2024) | 0.250 | 0.357 | 0.024 | 0.100 | 0.128 | 0.242 | 0.014 | 0.076 | 0.205 | 0.311 | 0.323 | 0.153 |

## E  MAIN RESULTS IN DETAILED

We reported detailed results for various baselines and different variants of our model across ten datasets, with missing rates ranging from 0.1 in Table 15 to 0.9 in Table 23.

Table 15: Detailed Imputation results with missing rate set to 0.1. We set the input length to 96. A lower MSE or MAE indicates a better imputation performance.

| Model | ETTh1 MSE | ETTh1 MAE | ETTh2 MSE | ETTh2 MAE | ETTm1 MSE | ETTm1 MAE | ETTm2 MSE | ETTm2 MAE | ECL MSE | ECL MAE | Weather MSE | Weather MAE | PEMS03 MSE | PEMS03 MAE | PEMS04 MSE | PEMS04 MAE | PEMS07 MSE | PEMS07 MAE | PEMS08 MSE | PEMS08 MAE |
|---|---|---|---|---|---|---|---|---|---|---|---|---|---|---|---|---|---|---|---|---|
| Median | 0.723 | 0.609 | 0.725 | 0.472 | 0.698 | 0.582 | 0.746 | 0.464 | 0.992 | 0.825 | 0.991 | 0.494 | 0.667 | 0.597 | 0.716 | 0.633 | 0.728 | 0.634 | 0.708 | 0.641 |
| Last | 0.399 | 0.460 | 0.068 | 0.135 | 0.310 | 0.399 | 0.039 | 0.109 | 0.912 | 0.811 | 0.692 | 0.399 | 0.448 | 0.489 | 0.460 | 0.501 | 0.480 | 0.500 | 0.420 | 0.479 |
| Autoformer(2021) | 0.487 | 0.529 | 0.629 | 0.509 | 0.518 | 0.477 | 0.231 | 0.319 | 0.074 | 0.194 | 0.435 | 0.370 | 2.698 | 1.348 | 0.532 | 0.583 | 0.677 | 0.670 | 3.714 | 1.575 |
| Fedformer(2022) | 0.276 | 0.393 | 1.278 | 0.645 | 0.030 | 0.120 | 0.023 | 0.106 | 0.072 | 0.193 | 0.142 | 0.126 | 0.151 | 0.281 | 0.455 | 0.503 | 0.361 | 0.463 | 0.252 | 0.334 |
| Dlinear(2023) | 0.284 | 0.383 | 0.235 | 0.330 | 0.262 | 0.383 | 0.399 | 0.472 | 0.259 | 0.403 | 0.355 | 0.353 | 0.204 | 0.377 | 0.219 | 0.388 | 0.283 | 0.445 | 0.315 | 0.468 |
| iTransformer(2024) | 0.447 | 0.490 | 0.056 | 0.147 | 0.129 | 0.254 | 0.074 | 0.196 | 0.053 | 0.153 | 0.308 | 0.243 | 0.075 | 0.192 | 0.084 | 0.204 | 0.062 | 0.175 | 0.097 | 0.220 |
| BRITS(2018) | 0.119 | 0.226 | 0.063 | 0.127 | 0.046 | 0.122 | 0.032 | 0.084 | 0.233 | 0.367 | 0.798 | 0.476 | 0.140 | 0.263 | 0.252 | 0.362 | 0.227 | 0.353 | 0.210 | 0.331 |
| TimesNet(2022) | 0.109 | 0.230 | 0.014 | 0.080 | 0.034 | 0.118 | 0.008 | 0.056 | 0.319 | 0.422 | 0.617 | 0.271 | 0.086 | 0.200 | 0.139 | 0.253 | 0.119 | 0.236 | 0.100 | 0.213 |
| PatchTST(2023) | 0.123 | 0.246 | 0.017 | 0.086 | 0.059 | 0.167 | 0.009 | 0.061 | 0.078 | 0.216 | 0.193 | 0.105 | 0.051 | 0.166 | 0.070 | 0.192 | 0.044 | 0.151 | 0.064 | 0.184 |
| SAITS(2023) | 0.124 | 0.231 | 0.188 | 0.176 | 0.064 | 0.130 | 0.135 | 0.139 | 0.392 | 0.480 | 0.930 | 0.492 | 0.154 | 0.282 | 0.264 | 0.374 | 0.223 | 0.346 | 0.245 | 0.359 |
| GPT4TS(2024) | 0.095 | 0.207 | 0.017 | 0.079 | 0.028 | 0.103 | 0.008 | 0.053 | 0.223 | 0.348 | 0.968 | 0.317 | 0.087 | 0.205 | 0.143 | 0.256 | 0.116 | 0.234 | 0.096 | 0.215 |
| NuwaTS(specific) | 0.123 | 0.244 | 0.017 | 0.086 | 0.052 | 0.152 | 0.009 | 0.062 | 0.047 | 0.147 | 0.237 | 0.112 | 0.040 | 0.138 | 0.050 | 0.148 | 0.033 | 0.119 | 0.054 | 0.156 |
| PatchTST(one-for-all) | 0.116 | 0.237 | 0.014 | 0.080 | 0.044 | 0.138 | 0.009 | 0.060 | 0.072 | 0.199 | 0.175 | 0.100 | 0.047 | 0.156 | 0.057 | 0.169 | 0.041 | 0.146 | 0.051 | 0.158 |
| NuwaTS(one-for-all) | 0.101 | 0.209 | 0.014 | 0.075 | 0.034 | 0.113 | 0.008 | 0.055 | 0.046 | 0.142 | 0.141 | 0.075 | 0.040 | 0.138 | 0.052 | 0.152 | 0.033 | 0.121 | 0.046 | 0.139 |
| NuwaTS(fine-tuned) | 0.100 | 0.208 | 0.014 | 0.076 | 0.035 | 0.114 | 0.008 | 0.054 | 0.045 | 0.141 | 0.138 | 0.072 | 0.041 | 0.139 | 0.053 | 0.155 | 0.034 | 0.123 | 0.046 | 0.140 |

Table 16: Detailed Imputation results with missing rate set to 0.2. We set the input length to 96. A lower MSE or MAE indicates a better imputation performance.

| Model | ETTh1 | | ETTh2 | | ETTm1 | | ETTm2 | | ECL | | Weather | | PEMS03 | | PEMS04 | | PEMS07 | | PEMS08 | |
|---|---|---|---|---|---|---|---|---|---|---|---|---|---|---|---|---|---|---|---|---|
| | MSE | MAE | MSE | MAE | MSE | MAE | MSE | MAE | MSE | MAE | MSE | MAE | MSE | MAE | MSE | MAE | MSE | MAE | MSE | MAE |
| Median | 0.713 | 0.606 | 0.725 | 0.471 | 0.692 | 0.580 | 0.741 | 0.462 | 0.994 | 0.825 | 0.997 | 0.492 | 0.679 | 0.603 | 0.727 | 0.638 | 0.740 | 0.639 | 0.719 | 0.646 |
| Last | 0.402 | 0.461 | 0.071 | 0.136 | 0.312 | 0.400 | 0.041 | 0.110 | 0.917 | 0.813 | 0.703 | 0.340 | 0.450 | 0.490 | 0.463 | 0.502 | 0.482 | 0.502 | 0.422 | 0.480 |
| Autoformer(2021) | 0.425 | 0.493 | 0.532 | 0.472 | 0.345 | 0.407 | 0.139 | 0.263 | 0.082 | 0.204 | 0.423 | 0.342 | 1.168 | 0.880 | 0.218 | 0.356 | 0.250 | 0.384 | 1.725 | 1.061 |
| Fedformer(2022) | 0.270 | 0.388 | 0.806 | 0.536 | 0.043 | 0.145 | 0.026 | 0.113 | 0.080 | 0.204 | 0.162 | 0.138 | 0.151 | 0.285 | 0.357 | 0.445 | 0.293 | 0.410 | 0.223 | 0.323 |
| Dlinear(2023) | 0.215 | 0.331 | 0.120 | 0.227 | 0.153 | 0.284 | 0.219 | 0.345 | 0.179 | 0.323 | 0.263 | 0.253 | 0.114 | 0.266 | 0.126 | 0.280 | 0.156 | 0.319 | 0.178 | 0.341 |
| iTransformer(2024) | 0.480 | 0.507 | 0.101 | 0.199 | 0.154 | 0.279 | 0.104 | 0.235 | 0.058 | 0.160 | 0.344 | 0.277 | 0.080 | 0.199 | 0.091 | 0.216 | 0.068 | 0.186 | 0.107 | 0.235 |
| BRITS(2018) | 0.131 | 0.241 | 0.069 | 0.133 | 0.050 | 0.128 | 0.036 | 0.089 | 0.243 | 0.375 | 0.779 | 0.474 | 0.140 | 0.263 | 0.252 | 0.363 | 0.226 | 0.352 | 0.212 | 0.332 |
| TimesNet(2022) | 0.114 | 0.235 | 0.015 | 0.081 | 0.036 | 0.120 | 0.008 | 0.057 | 0.323 | 0.425 | 0.635 | 0.272 | 0.088 | 0.203 | 0.140 | 0.255 | 0.120 | 0.238 | 0.102 | 0.215 |
| PatchTST(2023) | 0.131 | 0.253 | 0.018 | 0.087 | 0.059 | 0.164 | 0.009 | 0.061 | 0.082 | 0.219 | 0.198 | 0.107 | 0.050 | 0.162 | 0.068 | 0.187 | 0.043 | 0.150 | 0.062 | 0.177 |
| SAITS(2023) | 0.131 | 0.238 | 0.177 | 0.169 | 0.063 | 0.131 | 0.132 | 0.125 | 0.405 | 0.488 | 0.935 | 0.492 | 0.154 | 0.283 | 0.264 | 0.375 | 0.224 | 0.346 | 0.245 | 0.360 |
| GPT4TS(2024) | 0.111 | 0.221 | 0.018 | 0.081 | 0.032 | 0.110 | 0.009 | 0.055 | 0.233 | 0.356 | 0.973 | 0.317 | 0.088 | 0.206 | 0.144 | 0.258 | 0.118 | 0.236 | 0.096 | 0.213 |
| NuwaTS(specific) | 0.129 | 0.250 | 0.017 | 0.086 | 0.049 | 0.144 | 0.009 | 0.060 | 0.049 | 0.149 | 0.241 | 0.114 | 0.040 | 0.137 | 0.049 | 0.147 | 0.032 | 0.118 | 0.051 | 0.150 |
| PatchTST(one-for-all) | 0.120 | 0.240 | 0.015 | 0.080 | 0.045 | 0.139 | 0.009 | 0.060 | 0.075 | 0.202 | 0.177 | 0.100 | 0.047 | 0.157 | 0.057 | 0.169 | 0.041 | 0.146 | 0.051 | 0.157 |
| NuwaTS(one-for-all) | 0.103 | 0.209 | 0.014 | 0.075 | 0.035 | 0.111 | 0.008 | 0.054 | 0.049 | 0.145 | 0.148 | 0.072 | 0.038 | 0.135 | 0.049 | 0.148 | 0.032 | 0.118 | 0.044 | 0.136 |
| NuwaTS(fine-tuned) | 0.101 | 0.207 | 0.014 | 0.075 | 0.035 | 0.111 | 0.008 | 0.053 | 0.047 | 0.144 | 0.140 | 0.071 | 0.039 | 0.135 | 0.050 | 0.149 | 0.032 | 0.118 | 0.044 | 0.136 |

Table 17: Detailed Imputation results with missing rate set to 0.3. We set the input length to 96. A lower MSE or MAE indicates a better imputation performance.

| Model | ETTh1 | | ETTh2 | | ETTm1 | | ETTm2 | | ECL | | Weather | | PEMS03 | | PEMS04 | | PEMS07 | | PEMS08 | |
|---|---|---|---|---|---|---|---|---|---|---|---|---|---|---|---|---|---|---|---|---|
| | MSE | MAE | MSE | MAE | MSE | MAE | MSE | MAE | MSE | MAE | MSE | MAE | MSE | MAE | MSE | MAE | MSE | MAE | MSE | MAE |
| Median | 0.707 | 0.605 | 0.717 | 0.469 | 0.689 | 0.579 | 0.740 | 0.462 | 0.988 | 0.826 | 0.991 | 0.492 | 0.681 | 0.604 | 0.730 | 0.639 | 0.742 | 0.640 | 0.722 | 0.647 |
| Last | 0.406 | 0.463 | 0.072 | 0.138 | 0.314 | 0.401 | 0.043 | 0.111 | 0.922 | 0.814 | 0.703 | 0.341 | 0.452 | 0.492 | 0.465 | 0.504 | 0.485 | 0.504 | 0.425 | 0.482 |
| Autoformer(2021) | 0.409 | 0.481 | 0.423 | 0.418 | 0.279 | 0.373 | 0.120 | 0.248 | 0.091 | 0.217 | 0.444 | 0.359 | 0.664 | 0.644 | 0.190 | 0.336 | 0.188 | 0.334 | 1.012 | 0.790 |
| Fedformer(2022) | 0.271 | 0.388 | 0.574 | 0.468 | 0.038 | 0.136 | 0.029 | 0.121 | 0.088 | 0.216 | 0.162 | 0.149 | 0.157 | 0.294 | 0.296 | 0.405 | 0.248 | 0.374 | 0.206 | 0.319 |
| Dlinear(2023) | 0.187 | 0.307 | 0.061 | 0.159 | 0.098 | 0.221 | 0.112 | 0.243 | 0.143 | 0.279 | 0.200 | 0.181 | 0.074 | 0.202 | 0.083 | 0.214 | 0.088 | 0.227 | 0.103 | 0.246 |
| iTransformer(2024) | 0.517 | 0.523 | 0.155 | 0.250 | 0.186 | 0.307 | 0.147 | 0.281 | 0.062 | 0.167 | 0.366 | 0.313 | 0.086 | 0.209 | 0.100 | 0.227 | 0.075 | 0.196 | 0.120 | 0.250 |
| BRITS(2018) | 0.143 | 0.254 | 0.076 | 0.138 | 0.055 | 0.135 | 0.040 | 0.094 | 0.254 | 0.384 | 0.783 | 0.474 | 0.140 | 0.263 | 0.253 | 0.363 | 0.225 | 0.351 | 0.214 | 0.334 |
| TimesNet(2022) | 0.120 | 0.241 | 0.017 | 0.083 | 0.038 | 0.123 | 0.009 | 0.057 | 0.329 | 0.428 | 0.652 | 0.273 | 0.090 | 0.206 | 0.142 | 0.257 | 0.121 | 0.239 | 0.103 | 0.217 |
| PatchTST(2023) | 0.138 | 0.259 | 0.018 | 0.088 | 0.059 | 0.163 | 0.009 | 0.062 | 0.086 | 0.222 | 0.213 | 0.108 | 0.050 | 0.160 | 0.066 | 0.183 | 0.043 | 0.149 | 0.060 | 0.173 |
| SAITS(2023) | 0.139 | 0.246 | 0.180 | 0.169 | 0.064 | 0.135 | 0.132 | 0.124 | 0.417 | 0.495 | 0.947 | 0.490 | 0.155 | 0.283 | 0.266 | 0.376 | 0.224 | 0.347 | 0.246 | 0.360 |
| GPT4TS(2024) | 0.124 | 0.235 | 0.019 | 0.083 | 0.038 | 0.118 | 0.009 | 0.056 | 0.244 | 0.364 | 0.953 | 0.317 | 0.089 | 0.207 | 0.146 | 0.260 | 0.119 | 0.238 | 0.097 | 0.212 |
| NuwaTS(specific) | 0.136 | 0.257 | 0.017 | 0.086 | 0.048 | 0.140 | 0.009 | 0.059 | 0.052 | 0.153 | 0.251 | 0.115 | 0.040 | 0.138 | 0.049 | 0.148 | 0.032 | 0.118 | 0.049 | 0.146 |
| PatchTST(one-for-all) | 0.126 | 0.245 | 0.015 | 0.080 | 0.047 | 0.142 | 0.009 | 0.060 | 0.079 | 0.206 | 0.174 | 0.101 | 0.047 | 0.157 | 0.057 | 0.170 | 0.042 | 0.147 | 0.052 | 0.158 |
| NuwaTS(one-for-all) | 0.109 | 0.214 | 0.014 | 0.076 | 0.036 | 0.113 | 0.008 | 0.054 | 0.052 | 0.150 | 0.152 | 0.073 | 0.038 | 0.134 | 0.049 | 0.148 | 0.032 | 0.118 | 0.044 | 0.135 |
| NuwaTS(fine-tuned) | 0.106 | 0.211 | 0.014 | 0.075 | 0.036 | 0.113 | 0.008 | 0.053 | 0.050 | 0.148 | 0.146 | 0.071 | 0.038 | 0.134 | 0.049 | 0.148 | 0.032 | 0.118 | 0.044 | 0.135 |

Table 18: Detailed Imputation results with missing rate set to 0.4. We set the input length to 96. A lower MSE or MAE indicates a better imputation performance.

| Model | ETTh1 | | ETTh2 | | ETTm1 | | ETTm2 | | ECL | | Weather | | PEMS03 | | PEMS04 | | PEMS07 | | PEMS08 | |
|---|---|---|---|---|---|---|---|---|---|---|---|---|---|---|---|---|---|---|---|---|
| | MSE | MAE | MSE | MAE | MSE | MAE | MSE | MAE | MSE | MAE | MSE | MAE | MSE | MAE | MSE | MAE | MSE | MAE | MSE | MAE |
| Median | 0.710 | 0.606 | 0.716 | 0.469 | 0.687 | 0.579 | 0.736 | 0.461 | 0.988 | 0.827 | 0.984 | 0.493 | 0.688 | 0.607 | 0.736 | 0.642 | 0.749 | 0.643 | 0.728 | 0.650 |
| Last | 0.411 | 0.466 | 0.073 | 0.139 | 0.318 | 0.403 | 0.045 | 0.113 | 0.930 | 0.816 | 0.701 | 0.343 | 0.455 | 0.494 | 0.468 | 0.506 | 0.488 | 0.506 | 0.428 | 0.484 |
| Autoformer(2021) | 0.426 | 0.487 | 0.347 | 0.367 | 0.238 | 0.349 | 0.113 | 0.239 | 0.103 | 0.232 | 0.501 | 0.393 | 0.419 | 0.500 | 0.231 | 0.374 | 0.213 | 0.356 | 0.632 | 0.608 |
| Fedformer(2022) | 0.280 | 0.395 | 0.447 | 0.421 | 0.053 | 0.164 | 0.033 | 0.129 | 0.100 | 0.231 | 0.174 | 0.163 | 0.171 | 0.310 | 0.255 | 0.378 | 0.217 | 0.350 | 0.209 | 0.321 |
| Dlinear(2023) | 0.194 | 0.311 | 0.046 | 0.138 | 0.084 | 0.200 | 0.055 | 0.167 | 0.144 | 0.275 | 0.182 | 0.149 | 0.073 | 0.200 | 0.080 | 0.207 | 0.064 | 0.185 | 0.073 | 0.195 |
| iTransformer(2024) | 0.565 | 0.542 | 0.230 | 0.309 | 0.235 | 0.345 | 0.213 | 0.341 | 0.069 | 0.175 | 0.412 | 0.357 | 0.094 | 0.221 | 0.110 | 0.240 | 0.083 | 0.208 | 0.135 | 0.266 |
| BRITS(2018) | 0.162 | 0.273 | 0.086 | 0.147 | 0.062 | 0.145 | 0.047 | 0.101 | 0.270 | 0.395 | 0.781 | 0.472 | 0.140 | 0.264 | 0.253 | 0.364 | 0.224 | 0.350 | 0.217 | 0.336 |
| TimesNet(2022) | 0.128 | 0.249 | 0.017 | 0.085 | 0.041 | 0.128 | 0.009 | 0.059 | 0.336 | 0.433 | 0.664 | 0.274 | 0.093 | 0.210 | 0.144 | 0.259 | 0.123 | 0.242 | 0.106 | 0.221 |
| PatchTST(2023) | 0.148 | 0.269 | 0.019 | 0.089 | 0.061 | 0.163 | 0.009 | 0.062 | 0.093 | 0.227 | 0.218 | 0.110 | 0.050 | 0.160 | 0.064 | 0.180 | 0.044 | 0.148 | 0.058 | 0.169 |
| SAITS(2023) | 0.150 | 0.257 | 0.191 | 0.174 | 0.066 | 0.139 | 0.132 | 0.126 | 0.429 | 0.503 | 0.936 | 0.486 | 0.155 | 0.284 | 0.267 | 0.377 | 0.226 | 0.347 | 0.246 | 0.361 |
| GPT4TS(2024) | 0.143 | 0.252 | 0.021 | 0.085 | 0.045 | 0.128 | 0.010 | 0.058 | 0.258 | 0.375 | 0.945 | 0.317 | 0.090 | 0.208 | 0.149 | 0.262 | 0.122 | 0.240 | 0.099 | 0.214 |
| NuwaTS(specific) | 0.145 | 0.265 | 0.017 | 0.086 | 0.050 | 0.140 | 0.009 | 0.059 | 0.058 | 0.160 | 0.264 | 0.117 | 0.041 | 0.139 | 0.050 | 0.149 | 0.033 | 0.119 | 0.049 | 0.145 |
| PatchTST(one-for-all) | 0.136 | 0.254 | 0.016 | 0.082 | 0.051 | 0.147 | 0.009 | 0.060 | 0.085 | 0.212 | 0.187 | 0.104 | 0.048 | 0.159 | 0.058 | 0.171 | 0.043 | 0.148 | 0.052 | 0.159 |
| NuwaTS(one-for-all) | 0.119 | 0.225 | 0.015 | 0.077 | 0.040 | 0.118 | 0.009 | 0.055 | 0.057 | 0.157 | 0.162 | 0.075 | 0.039 | 0.135 | 0.050 | 0.148 | 0.033 | 0.119 | 0.044 | 0.136 |
| NuwaTS(fine-tuned) | 0.115 | 0.220 | 0.014 | 0.076 | 0.040 | 0.117 | 0.008 | 0.054 | 0.055 | 0.155 | 0.168 | 0.074 | 0.039 | 0.135 | 0.050 | 0.148 | 0.033 | 0.119 | 0.044 | 0.136 |

Table 19: Detailed Imputation results with missing rate set to 0.5. We set the input length to 96. A lower MSE or MAE indicates a better imputation performance.

| Model | ETTh1 | | ETTh2 | | ETTm1 | | ETTm2 | | ECL | | Weather | | PEMS03 | | PEMS04 | | PEMS07 | | PEMS08 | |
| --- | --- | --- | --- | --- | --- | --- | --- | --- | --- | --- | --- | --- | --- | --- | --- | --- | --- | --- | --- | --- |
| | MSE | MAE | MSE | MAE | MSE | MAE | MSE | MAE | MSE | MAE | MSE | MAE | MSE | MAE | MSE | MAE | MSE | MAE | MSE | MAE |
| Median | 0.708 | 0.605 | 0.721 | 0.470 | 0.686 | 0.578 | 0.735 | 0.461 | 0.994 | 0.830 | 0.991 | 0.493 | 0.669 | 0.597 | 0.721 | 0.634 | 0.731 | 0.634 | 0.712 | 0.642 |
| Last | 0.416 | 0.469 | 0.078 | 0.142 | 0.323 | 0.406 | 0.049 | 0.115 | 0.941 | 0.820 | 0.716 | 0.345 | 0.460 | 0.498 | 0.474 | 0.509 | 0.493 | 0.509 | 0.433 | 0.487 |
| Autoformer(2021) | 0.469 | 0.507 | 0.320 | 0.336 | 0.223 | 0.339 | 0.116 | 0.237 | 0.119 | 0.249 | 0.571 | 0.432 | 0.329 | 0.441 | 0.301 | 0.432 | 0.276 | 0.408 | 0.461 | 0.515 |
| Fedformer(2022) | 0.298 | 0.407 | 0.389 | 0.394 | 0.061 | 0.176 | 0.038 | 0.139 | 0.114 | 0.248 | 0.192 | 0.182 | 0.197 | 0.336 | 0.241 | 0.371 | 0.210 | 0.346 | 0.203 | 0.331 |
| Dlinear(2023) | 0.239 | 0.345 | 0.084 | 0.183 | 0.122 | 0.240 | 0.062 | 0.177 | 0.189 | 0.319 | 0.212 | 0.186 | 0.120 | 0.268 | 0.125 | 0.272 | 0.094 | 0.233 | 0.099 | 0.234 |
| iTransformer(2024) | 0.620 | 0.564 | 0.321 | 0.372 | 0.297 | 0.390 | 0.299 | 0.405 | 0.077 | 0.187 | 0.473 | 0.405 | 0.104 | 0.234 | 0.122 | 0.253 | 0.093 | 0.220 | 0.151 | 0.283 |
| BRITS(2018) | 0.184 | 0.294 | 0.097 | 0.157 | 0.072 | 0.158 | 0.055 | 0.109 | 0.290 | 0.410 | 0.790 | 0.471 | 0.141 | 0.264 | 0.255 | 0.366 | 0.223 | 0.349 | 0.221 | 0.339 |
| TimesNet(2022) | 0.140 | 0.260 | 0.019 | 0.087 | 0.046 | 0.135 | 0.010 | 0.060 | 0.345 | 0.439 | 0.683 | 0.276 | 0.146 | 0.262 | 0.126 | 0.245 | 0.109 | 0.225 | 0.118 | 0.230 |
| PatchTST(2023) | 0.162 | 0.281 | 0.020 | 0.090 | 0.065 | 0.166 | 0.010 | 0.063 | 0.102 | 0.236 | 0.239 | 0.113 | 0.051 | 0.161 | 0.064 | 0.178 | 0.045 | 0.149 | 0.059 | 0.168 |
| SAITS(2023) | 0.166 | 0.271 | 0.203 | 0.180 | 0.070 | 0.146 | 0.134 | 0.131 | 0.441 | 0.510 | 0.935 | 0.483 | 0.156 | 0.284 | 0.269 | 0.379 | 0.227 | 0.348 | 0.248 | 0.362 |
| GPT4TS(2024) | 0.169 | 0.273 | 0.022 | 0.088 | 0.056 | 0.141 | 0.011 | 0.060 | 0.275 | 0.387 | 0.939 | 0.317 | 0.093 | 0.211 | 0.152 | 0.265 | 0.124 | 0.243 | 0.102 | 0.217 |
| NuwaTS(specific) | 0.158 | 0.277 | 0.018 | 0.087 | 0.054 | 0.144 | 0.009 | 0.060 | 0.065 | 0.171 | 0.280 | 0.120 | 0.043 | 0.141 | 0.052 | 0.151 | 0.035 | 0.122 | 0.049 | 0.146 |
| PatchTST(one-for-all) | 0.150 | 0.267 | 0.017 | 0.084 | 0.057 | 0.155 | 0.010 | 0.062 | 0.093 | 0.221 | 0.195 | 0.107 | 0.050 | 0.161 | 0.060 | 0.172 | 0.044 | 0.150 | 0.054 | 0.161 |
| NuwaTS(one-for-all) | 0.135 | 0.241 | 0.016 | 0.080 | 0.045 | 0.126 | 0.009 | 0.056 | 0.064 | 0.167 | 0.180 | 0.079 | 0.041 | 0.138 | 0.051 | 0.150 | 0.034 | 0.121 | 0.046 | 0.138 |
| NuwaTS(fine-tuned) | 0.128 | 0.233 | 0.015 | 0.078 | 0.045 | 0.124 | 0.009 | 0.055 | 0.062 | 0.164 | 0.188 | 0.077 | 0.041 | 0.138 | 0.051 | 0.150 | 0.034 | 0.121 | 0.046 | 0.138 |

Table 20: Detailed Imputation results with missing rate set to 0.6. We set the input length to 96. A lower MSE or MAE indicates a better imputation performance.

| Model | ETTh1 | | ETTh2 | | ETTm1 | | ETTm2 | | ECL | | Weather | | PEMS03 | | PEMS04 | | PEMS07 | | PEMS08 | |
| --- | --- | --- | --- | --- | --- | --- | --- | --- | --- | --- | --- | --- | --- | --- | --- | --- | --- | --- | --- | --- |
| | MSE | MAE | MSE | MAE | MSE | MAE | MSE | MAE | MSE | MAE | MSE | MAE | MSE | MAE | MSE | MAE | MSE | MAE | MSE | MAE |
| Median | 0.710 | 0.607 | 0.718 | 0.470 | 0.689 | 0.580 | 0.736 | 0.461 | 1.005 | 0.829 | 0.990 | 0.495 | 0.686 | 0.606 | 0.735 | 0.642 | 0.747 | 0.642 | 0.728 | 0.650 |
| Last | 0.425 | 0.473 | 0.083 | 0.145 | 0.329 | 0.409 | 0.054 | 0.119 | 0.955 | 0.824 | 0.725 | 0.347 | 0.467 | 0.502 | 0.480 | 0.513 | 0.499 | 0.513 | 0.439 | 0.491 |
| Autoformer(2021) | 0.526 | 0.535 | 0.335 | 0.335 | 0.234 | 0.347 | 0.136 | 0.254 | 0.137 | 0.269 | 0.640 | 0.469 | 0.322 | 0.439 | 0.378 | 0.489 | 0.349 | 0.463 | 0.408 | 0.487 |
| Fedformer(2022) | 0.323 | 0.423 | 0.348 | 0.370 | 0.060 | 0.172 | 0.045 | 0.151 | 0.131 | 0.266 | 0.216 | 0.204 | 0.235 | 0.369 | 0.254 | 0.383 | 0.230 | 0.362 | 0.218 | 0.346 |
| Dlinear(2023) | 0.313 | 0.397 | 0.163 | 0.261 | 0.198 | 0.313 | 0.125 | 0.254 | 0.265 | 0.390 | 0.279 | 0.262 | 0.203 | 0.364 | 0.206 | 0.365 | 0.169 | 0.327 | 0.170 | 0.324 |
| iTransformer(2024) | 0.674 | 0.585 | 0.418 | 0.430 | 0.367 | 0.437 | 0.394 | 0.467 | 0.087 | 0.198 | 0.539 | 0.450 | 0.113 | 0.245 | 0.135 | 0.267 | 0.103 | 0.232 | 0.166 | 0.297 |
| BRITS(2018) | 0.210 | 0.318 | 0.112 | 0.169 | 0.085 | 0.175 | 0.066 | 0.119 | 0.314 | 0.427 | 0.791 | 0.471 | 0.141 | 0.265 | 0.257 | 0.369 | 0.224 | 0.348 | 0.225 | 0.343 |
| TimesNet(2022) | 0.155 | 0.273 | 0.020 | 0.090 | 0.053 | 0.144 | 0.010 | 0.062 | 0.356 | 0.446 | 0.696 | 0.277 | 0.100 | 0.219 | 0.149 | 0.266 | 0.129 | 0.249 | 0.129 | 0.230 |
| PatchTST(2023) | 0.180 | 0.297 | 0.021 | 0.092 | 0.071 | 0.173 | 0.010 | 0.065 | 0.115 | 0.249 | 0.259 | 0.118 | 0.053 | 0.164 | 0.066 | 0.180 | 0.047 | 0.152 | 0.060 | 0.170 |
| SAITS(2023) | 0.185 | 0.287 | 0.217 | 0.188 | 0.076 | 0.156 | 0.135 | 0.138 | 0.453 | 0.517 | 0.934 | 0.480 | 0.156 | 0.285 | 0.271 | 0.380 | 0.228 | 0.349 | 0.249 | 0.364 |
| GPT4TS(2024) | 0.197 | 0.295 | 0.025 | 0.092 | 0.069 | 0.157 | 0.012 | 0.063 | 0.293 | 0.401 | 0.924 | 0.318 | 0.096 | 0.215 | 0.155 | 0.269 | 0.128 | 0.247 | 0.106 | 0.222 |
| NuwaTS(specific) | 0.175 | 0.291 | 0.019 | 0.088 | 0.060 | 0.151 | 0.010 | 0.062 | 0.075 | 0.183 | 0.297 | 0.123 | 0.045 | 0.145 | 0.054 | 0.154 | 0.037 | 0.125 | 0.051 | 0.148 |
| PatchTST(one-for-all) | 0.169 | 0.284 | 0.018 | 0.086 | 0.065 | 0.165 | 0.010 | 0.063 | 0.104 | 0.232 | 0.209 | 0.111 | 0.052 | 0.163 | 0.062 | 0.175 | 0.046 | 0.152 | 0.056 | 0.163 |
| NuwaTS(one-for-all) | 0.155 | 0.260 | 0.017 | 0.083 | 0.053 | 0.137 | 0.010 | 0.058 | 0.073 | 0.179 | 0.200 | 0.084 | 0.043 | 0.141 | 0.054 | 0.154 | 0.036 | 0.124 | 0.048 | 0.142 |
| NuwaTS(fine-tuned) | 0.146 | 0.249 | 0.016 | 0.080 | 0.051 | 0.134 | 0.010 | 0.057 | 0.070 | 0.176 | 0.202 | 0.082 | 0.043 | 0.141 | 0.053 | 0.153 | 0.036 | 0.124 | 0.048 | 0.141 |

Table 21: Detailed Imputation results with missing rate set to 0.7. We set the input length to 96. A lower MSE or MAE indicates a better imputation performance.

| Model | ETTh1 | | ETTh2 | | ETTm1 | | ETTm2 | | ECL | | Weather | | PEMS03 | | PEMS04 | | PEMS07 | | PEMS08 | |
| --- | --- | --- | --- | --- | --- | --- | --- | --- | --- | --- | --- | --- | --- | --- | --- | --- | --- | --- | --- | --- |
| | MSE | MAE | MSE | MAE | MSE | MAE | MSE | MAE | MSE | MAE | MSE | MAE | MSE | MAE | MSE | MAE | MSE | MAE | MSE | MAE |
| Median | 0.717 | 0.610 | 0.718 | 0.470 | 0.694 | 0.583 | 0.744 | 0.463 | 1.015 | 0.833 | 1.002 | 0.499 | 0.710 | 0.623 | 0.758 | 0.657 | 0.775 | 0.658 | 0.750 | 0.665 |
| Last | 0.438 | 0.480 | 0.093 | 0.151 | 0.341 | 0.416 | 0.062 | 0.125 | 0.980 | 0.830 | 0.740 | 0.351 | 0.478 | 0.509 | 0.491 | 0.520 | 0.511 | 0.520 | 0.450 | 0.498 |
| Autoformer(2021) | 0.612 | 0.576 | 0.402 | 0.368 | 0.284 | 0.382 | 0.201 | 0.311 | 0.165 | 0.295 | 0.725 | 0.514 | 0.376 | 0.476 | 0.477 | 0.556 | 0.446 | 0.531 | 0.429 | 0.502 |
| Fedformer(2022) | 0.369 | 0.449 | 0.288 | 0.332 | 0.077 | 0.196 | 0.057 | 0.171 | 0.155 | 0.289 | 0.256 | 0.238 | 0.303 | 0.421 | 0.305 | 0.420 | 0.292 | 0.407 | 0.261 | 0.379 |
| Dlinear(2023) | 0.432 | 0.470 | 0.301 | 0.366 | 0.331 | 0.417 | 0.256 | 0.370 | 0.391 | 0.491 | 0.397 | 0.363 | 0.339 | 0.487 | 0.341 | 0.484 | 0.303 | 0.454 | 0.301 | 0.449 |
| iTransformer(2024) | 0.742 | 0.613 | 0.544 | 0.497 | 0.466 | 0.499 | 0.518 | 0.537 | 0.102 | 0.216 | 0.627 | 0.502 | 0.122 | 0.256 | 0.152 | 0.254 | 0.117 | 0.247 | 0.184 | 0.313 |
| BRITS(2018) | 0.251 | 0.353 | 0.135 | 0.189 | 0.107 | 0.203 | 0.082 | 0.136 | 0.351 | 0.452 | 0.793 | 0.471 | 0.143 | 0.267 | 0.262 | 0.373 | 0.226 | 0.350 | 0.232 | 0.349 |
| TimesNet(2022) | 0.180 | 0.296 | 0.023 | 0.094 | 0.066 | 0.160 | 0.011 | 0.065 | 0.374 | 0.458 | 0.710 | 0.280 | 0.105 | 0.227 | 0.155 | 0.273 | 0.135 | 0.257 | 0.118 | 0.237 |
| PatchTST(2023) | 0.208 | 0.320 | 0.023 | 0.095 | 0.082 | 0.186 | 0.011 | 0.067 | 0.139 | 0.273 | 0.292 | 0.126 | 0.058 | 0.170 | 0.070 | 0.185 | 0.051 | 0.159 | 0.065 | 0.176 |
| SAITS(2023) | 0.217 | 0.312 | 0.233 | 0.199 | 0.089 | 0.172 | 0.137 | 0.146 | 0.467 | 0.525 | 0.925 | 0.477 | 0.158 | 0.286 | 0.274 | 0.383 | 0.230 | 0.350 | 0.251 | 0.366 |
| GPT4TS(2024) | 0.239 | 0.328 | 0.028 | 0.097 | 0.092 | 0.182 | 0.013 | 0.067 | 0.320 | 0.420 | 0.918 | 0.320 | 0.102 | 0.223 | 0.161 | 0.275 | 0.134 | 0.254 | 0.112 | 0.229 |
| NuwaTS(specific) | 0.202 | 0.312 | 0.021 | 0.092 | 0.072 | 0.166 | 0.011 | 0.064 | 0.091 | 0.202 | 0.338 | 0.130 | 0.049 | 0.151 | 0.058 | 0.160 | 0.040 | 0.131 | 0.055 | 0.154 |
| PatchTST(one-for-all) | 0.199 | 0.308 | 0.020 | 0.091 | 0.079 | 0.182 | 0.011 | 0.066 | 0.122 | 0.249 | 0.232 | 0.118 | 0.056 | 0.168 | 0.066 | 0.179 | 0.049 | 0.157 | 0.060 | 0.168 |
| NuwaTS(one-for-all) | 0.188 | 0.289 | 0.019 | 0.087 | 0.067 | 0.157 | 0.011 | 0.061 | 0.090 | 0.198 | 0.229 | 0.092 | 0.047 | 0.147 | 0.058 | 0.159 | 0.040 | 0.130 | 0.052 | 0.147 |
| NuwaTS(fine-tuned) | 0.175 | 0.275 | 0.018 | 0.084 | 0.064 | 0.150 | 0.011 | 0.060 | 0.086 | 0.195 | 0.231 | 0.089 | 0.047 | 0.147 | 0.057 | 0.159 | 0.040 | 0.130 | 0.052 | 0.147 |

Table 22: Detailed Imputation results with missing rate set to 0.8. We set the input length to 96. A lower MSE or MAE indicates a better imputation performance.

| Model | ETTh1 | | ETTh2 | | ETTm1 | | ETTm2 | | ECL | | Weather | | PEMS03 | | PEMS04 | | PEMS07 | | PEMS08 | |
|---|---|---|---|---|---|---|---|---|---|---|---|---|---|---|---|---|---|---|---|---|
| | MSE | MAE | MSE | MAE | MSE | MAE | MSE | MAE | MSE | MAE | MSE | MAE | MSE | MAE | MSE | MAE | MSE | MAE | MSE | MAE |
| Median | 0.734 | 0.616 | 0.740 | 0.475 | 0.709 | 0.588 | 0.749 | 0.465 | 1.023 | 0.844 | 1.005 | 0.502 | 0.708 | 0.617 | 0.762 | 0.653 | 0.774 | 0.654 | 0.753 | 0.662 |
| Last | 0.463 | 0.491 | 0.110 | 0.162 | 0.362 | 0.427 | 0.077 | 0.134 | 1.019 | 0.840 | 0.766 | 0.358 | 0.498 | 0.521 | 0.511 | 0.532 | 0.531 | 0.532 | 0.469 | 0.510 |
| Autoformer(2021) | 0.724 | 0.624 | 0.523 | 0.432 | 0.381 | 0.443 | 0.311 | 0.398 | 0.200 | 0.324 | 0.811 | 0.557 | 0.477 | 0.540 | 0.581 | 0.620 | 0.551 | 0.598 | 0.514 | 0.552 |
| Fedformer(2022) | 0.448 | 0.491 | 0.264 | 0.304 | 0.123 | 0.248 | 0.078 | 0.198 | 0.185 | 0.315 | 0.327 | 0.295 | 0.403 | 0.490 | 0.398 | 0.483 | 0.399 | 0.481 | 0.348 | 0.441 |
| Dlinear(2023) | 0.574 | 0.544 | 0.480 | 0.469 | 0.499 | 0.521 | 0.465 | 0.506 | 0.536 | 0.586 | 0.557 | 0.469 | 0.502 | 0.605 | 0.502 | 0.598 | 0.470 | 0.576 | 0.468 | 0.571 |
| iTransformer(2024) | 0.810 | 0.642 | 0.671 | 0.557 | 0.581 | 0.562 | 0.647 | 0.602 | 0.127 | 0.242 | 0.720 | 0.552 | 0.128 | 0.262 | 0.170 | 0.300 | 0.132 | 0.263 | 0.208 | 0.334 |
| BRITS(2018) | 0.305 | 0.394 | 0.170 | 0.216 | 0.144 | 0.244 | 0.108 | 0.160 | 0.400 | 0.484 | 0.799 | 0.473 | 0.146 | 0.271 | 0.269 | 0.379 | 0.232 | 0.355 | 0.241 | 0.357 |
| TimesNet(2022) | 0.221 | 0.329 | 0.028 | 0.101 | 0.093 | 0.191 | 0.013 | 0.070 | 0.401 | 0.474 | 0.724 | 0.286 | 0.113 | 0.237 | 0.162 | 0.281 | 0.143 | 0.267 | 0.125 | 0.246 |
| PatchTST(2023) | 0.248 | 0.351 | 0.027 | 0.101 | 0.102 | 0.208 | 0.013 | 0.071 | 0.186 | 0.316 | 0.341 | 0.137 | 0.067 | 0.180 | 0.080 | 0.196 | 0.059 | 0.170 | 0.073 | 0.185 |
| SAITS(2023) | 0.269 | 0.347 | 0.253 | 0.214 | 0.111 | 0.198 | 0.139 | 0.155 | 0.484 | 0.534 | 0.924 | 0.477 | 0.160 | 0.289 | 0.277 | 0.385 | 0.232 | 0.352 | 0.255 | 0.369 |
| GPT4TS(2024) | 0.293 | 0.367 | 0.032 | 0.104 | 0.127 | 0.218 | 0.015 | 0.072 | 0.358 | 0.446 | 0.911 | 0.325 | 0.112 | 0.235 | 0.169 | 0.284 | 0.143 | 0.265 | 0.123 | 0.242 |
| NuwaTS(specific) | 0.242 | 0.343 | 0.024 | 0.097 | 0.092 | 0.189 | 0.013 | 0.068 | 0.119 | 0.231 | 0.372 | 0.139 | 0.057 | 0.160 | 0.066 | 0.169 | 0.047 | 0.141 | 0.062 | 0.164 |
| PatchTST(one-for-all) | 0.243 | 0.342 | 0.023 | 0.098 | 0.104 | 0.208 | 0.013 | 0.072 | 0.153 | 0.275 | 0.284 | 0.132 | 0.064 | 0.176 | 0.073 | 0.188 | 0.056 | 0.165 | 0.067 | 0.176 |
| NuwaTS(one-for-all) | 0.236 | 0.327 | 0.023 | 0.094 | 0.093 | 0.187 | 0.012 | 0.066 | 0.116 | 0.226 | 0.267 | 0.104 | 0.054 | 0.157 | 0.065 | 0.169 | 0.046 | 0.140 | 0.059 | 0.157 |
| NuwaTS(fine-tuned) | 0.218 | 0.310 | 0.021 | 0.090 | 0.085 | 0.175 | 0.012 | 0.064 | 0.112 | 0.223 | 0.269 | 0.100 | 0.054 | 0.156 | 0.064 | 0.168 | 0.046 | 0.139 | 0.058 | 0.156 |

Table 23: Detailed Imputation results with missing rate set to 0.9. We set the input length to 96. A lower MSE or MAE indicates a better imputation performance.

| Model | ETTh1 | | ETTh2 | | ETTm1 | | ETTm2 | | ECL | | Weather | | PEMS03 | | PEMS04 | | PEMS07 | | PEMS08 | |
|---|---|---|---|---|---|---|---|---|---|---|---|---|---|---|---|---|---|---|---|---|
| | MSE | MAE | MSE | MAE | MSE | MAE | MSE | MAE | MSE | MAE | MSE | MAE | MSE | MAE | MSE | MAE | MSE | MAE | MSE | MAE |
| Median | 0.786 | 0.635 | 0.768 | 0.483 | 0.748 | 0.603 | 0.772 | 0.472 | 1.086 | 0.870 | 1.027 | 0.513 | 0.733 | 0.626 | 0.795 | 0.665 | 0.805 | 0.667 | 0.788 | 0.676 |
| Last | 0.526 | 0.521 | 0.156 | 0.191 | 0.419 | 0.456 | 0.121 | 0.164 | 1.112 | 0.863 | 0.834 | 0.377 | 0.555 | 0.556 | 0.569 | 0.566 | 0.590 | 0.566 | 0.526 | 0.543 |
| Autoformer(2021) | 0.889 | 0.690 | 0.735 | 0.544 | 0.612 | 0.563 | 0.535 | 0.541 | 0.268 | 0.372 | 0.937 | 0.616 | 0.676 | 0.654 | 0.731 | 0.705 | 0.704 | 0.688 | 0.718 | 0.657 |
| Fedformer(2022) | 0.650 | 0.592 | 0.445 | 0.393 | 0.194 | 0.307 | 0.168 | 0.277 | 0.239 | 0.355 | 0.500 | 0.402 | 0.596 | 0.615 | 0.592 | 0.607 | 0.609 | 0.618 | 0.557 | 0.575 |
| Dlinear(2023) | 0.768 | 0.634 | 0.711 | 0.579 | 0.721 | 0.635 | 0.699 | 0.625 | 0.748 | 0.707 | 0.749 | 0.567 | 0.728 | 0.738 | 0.712 | 0.720 | 0.707 | 0.715 | 0.706 | 0.709 |
| iTransformer(2024) | 0.895 | 0.679 | 0.826 | 0.625 | 0.754 | 0.648 | 0.810 | 0.675 | 0.196 | 0.304 | 0.842 | 0.610 | 0.168 | 0.310 | 0.215 | 0.344 | 0.160 | 0.294 | 0.292 | 0.417 |
| BRITS(2018) | 0.415 | 0.464 | 0.244 | 0.277 | 0.245 | 0.336 | 0.171 | 0.218 | 0.502 | 0.548 | 0.825 | 0.483 | 0.155 | 0.282 | 0.281 | 0.389 | 0.246 | 0.369 | 0.258 | 0.371 |
| TimesNet(2022) | 0.328 | 0.407 | 0.039 | 0.115 | 0.191 | 0.278 | 0.018 | 0.081 | 0.478 | 0.520 | 0.750 | 0.307 | 0.145 | 0.269 | 0.188 | 0.306 | 0.170 | 0.293 | 0.152 | 0.274 |
| PatchTST(2023) | 0.328 | 0.404 | 0.038 | 0.114 | 0.159 | 0.260 | 0.018 | 0.082 | 0.344 | 0.436 | 0.482 | 0.172 | 0.098 | 0.211 | 0.116 | 0.230 | 0.094 | 0.205 | 0.104 | 0.216 |
| SAITS(2023) | 0.386 | 0.416 | 0.293 | 0.241 | 0.176 | 0.263 | 0.140 | 0.170 | 0.516 | 0.554 | 0.926 | 0.482 | 0.166 | 0.295 | 0.283 | 0.390 | 0.236 | 0.356 | 0.262 | 0.374 |
| GPT4TS(2024) | 0.390 | 0.431 | 0.043 | 0.118 | 0.213 | 0.294 | 0.020 | 0.083 | 0.460 | 0.511 | 0.925 | 0.338 | 0.148 | 0.271 | 0.200 | 0.313 | 0.176 | 0.296 | 0.158 | 0.278 |
| NuwaTS(specific) | 0.329 | 0.402 | 0.034 | 0.110 | 0.153 | 0.248 | 0.017 | 0.078 | 0.219 | 0.320 | 0.466 | 0.170 | 0.085 | 0.192 | 0.095 | 0.201 | 0.072 | 0.172 | 0.090 | 0.196 |
| PatchTST(one-for-all) | 0.344 | 0.410 | 0.034 | 0.113 | 0.179 | 0.275 | 0.019 | 0.086 | 0.308 | 0.389 | 0.440 | 0.173 | 0.093 | 0.206 | 0.105 | 0.218 | 0.085 | 0.196 | 0.094 | 0.204 |
| NuwaTS(one-for-all) | 0.331 | 0.394 | 0.032 | 0.109 | 0.172 | 0.264 | 0.018 | 0.080 | 0.219 | 0.309 | 0.377 | 0.141 | 0.083 | 0.189 | 0.095 | 0.201 | 0.074 | 0.174 | 0.086 | 0.188 |
| NuwaTS(fine-tuned) | 0.312 | 0.379 | 0.030 | 0.105 | 0.149 | 0.239 | 0.016 | 0.076 | 0.201 | 0.301 | 0.379 | 0.136 | 0.080 | 0.186 | 0.092 | 0.198 | 0.071 | 0.171 | 0.083 | 0.185 |

# F    LIMITATION

The model was trained on time series segments fixed at a length of 96, imputing the masked series based on the unmasked portions. Although our model can adapt to different domains and missing patterns, showing strong adaptability after domain-transfer fine-tuning, in practical applications, the model may require further fine-tuning when dealing with segments longer than 96, as well as the segments that are entirely missed.

Future work will aim to improve NuwaTS's capability in handling longer missing segments.

