# OpenReview forum: "NuwaTS: a Foundation Model Mending Every Incomplete Time Series"
_ICLR.cc/2025/Conference — ICLR 2025 Conference Withdrawn Submission_

### Official Review · Reviewer_nFYu · 2024-10-16

**Soundness:** 2
**Presentation:** 3
**Contribution:** 3
**Rating:** 5
**Confidence:** 3

**Summary:**

This paper proposes a single pre-trained model to imput on time series from different domains. Several designs are proposed such as statistics patch embedding, missingness embedding, time series path contrastive learning, and p-tuningv2 domain-specific fine-tuning. Experiments show that this one-for-all model achieves SOTA performance compared to strong baselines. Also, this model has zero-shot, few-shot and forecasting capability. Ablation studies show that the novel designs proposed are indeed helpful.

**Strengths:**

Experiments are conducted thoroughly.

Novel designs such as statistics patch embedding, missingnes embedding, time series path contrastive learning and p-tuningv2 style domain specific fine-tuning.

**Weaknesses:**

Though several designs are proposed, they are not as effective as the paper claims. In table 6, I don't think the improvements of these designs are significant enough.

Modeling each individual dimension separately is a limitation of the proposed method. Althought the authors proposes to include inter-series correlation information during fine-tuning for forecasting, they don't show that inter-series correlation information can be used for imputation. I think this method is not applicable to datasets with cross-dimensional interactions. Thus this is not a one-for-all model.

**Questions:**

None

---

### Official Review · Reviewer_m3f1 · 2024-11-04

**Soundness:** 3
**Presentation:** 2
**Contribution:** 3
**Rating:** 5
**Confidence:** 3

**Summary:**

This paper introduces NuwaTS, which is a general time series imputation framework leveraging pre-trained language models to address limitations in existing models' generalizability. Through tailored embeddings and contrastive learning, NuwaTS can impute missing data across any domain, with relatively small adjustments. The authors provide multiple experiments to show its performance over other models across diverse datasets.

**Strengths:**

- This paper is well written and organized. I enjoy reading this paper.
- I agree with the idea and appreciate the effort to deal with cross-domain generalization which is important in time-series modeling since the different patterns among domains are more significant than other data modalities.
- The authors provide well organized code base for the method which helps reproducibility and usability.
- Imputation-specific zero-shot model is new, and some designs(e.g. using missing information) can potentially beneficial for other tasks using time-series with missing data.

**Weaknesses:**

- Although this paper introduce several interesting concepts, I think the authors should justify their design choices since the authors integrates several existing components and conducts heavy engineering to boost the zero-shot capability, but there are sufficient techniques for existing works in time-series forecasting literature which can be extended to imputation task. I think the performance compared to other LLM based time series method is not that impressive
- I think proposing new benchmarking strategy(e.g. 1:1:1 split of train/valid/test in feature-wise manner) is insufficient to claim it as novel contribution. Since I think cross-variable generalization can be considered as part of domain generalization, how cross-variable generalization is specifically important? Just conducting zero-shot imputation experiments is not enough for measuring the generalization capabilities?
- This paper mainly lists the results of experiments. I think more insights and analysis beyond the abalation studies would be beneficial

**Questions:**

Please refer to weaknesses

---

### Official Review · Reviewer_gtH1 · 2024-11-04

**Soundness:** 3
**Presentation:** 3
**Contribution:** 2
**Rating:** 3
**Confidence:** 4

**Summary:**

This paper proposes to use pretrained language models (PLMs) for general times series imputation. This paper especially focused on the generalization on the variable dimension, within the domain and cross-domain. They proposed three different ways to pretrain / finetune PLMs with (optionally) few more parameters. They tested the performance on general imputation (missing at variable dimension), zero-shot domain transfer, few-shot finetuning, and downstream forecasting with TimesNet.

**Strengths:**

1. This paper includes three different settings to use PLMs for time series imputation. Different settings have their own challenges.
2. The paper includes comprehensive experimental results on diverse settings. Ablation studies are also comprehensive.

**Weaknesses:**

1. I couldn't find variances of the results. In my experience, the time series imputation performance often highly depends on the seed settings.
2. Related to 1: Table 4 zero-shot performance gain seems marginal compared to PatchTST. Please provide variances.
3. Table 8 results makes me doubt the model. For forecasting (which is an extreme case of imputation), the model seems worse (or just comparable) to baselines yet this uses much more parameters. I know the authors only trained small MLPs but it indeed uses large language model, which seems inefficient to me.
4. The baselines only includes predictive models.

**Questions:**

1. [Main] I am curious on the setting itself. I agree that sometimes missingnesses on variable-dimension is important. However, Is it indeed possible to impute new dimension which had never been exposed to the model? I assume this is only possible if the trend of the target variable is similar with the observed variables. In other words, the generalization performance indeed extremely rely on the variable splitting, which is randomly performed in this paper. Moreover, regarding marginal improvements on few-shot fine tuning results (table5), it seems that ETT dataset itself has high variable correlations. I think this paper must contain experiments or discussions with (synthetic) data which is trained with one pattern of variables then target variables have completely different missing patterns. I think this is the way to evaluate genuine generalization across variables and complete the claim of this paper. I would happy to dramatically raise my score if authors could show more careful analysis on across-variable generalization (and why this model is capable of those generalizability).
2. [Minor] Please provide variances of the results, and evaluation details such as number of iterations.
3. [Minor] Please include more recent probabilistic imputation models (VAE, GAN, DIffusion, and others..). If there is reason why those models are not applicable, please provide those reasons.

---

### Official Review · Reviewer_Hj8t · 2024-11-04

**Soundness:** 3
**Presentation:** 2
**Contribution:** 2
**Rating:** 3
**Confidence:** 4

**Summary:**

The paper introduces a framework called NuwaTS that utilizes pre-trained LMs as a backbone for performing time-series imputation. The paper employs normalization and embedding schemes from prior works, and introduces a novel way to represent missing values and domain-specific information as embeddings. These embedding schemes along with the pre-trained LMs are trained in an end-to-end manner to perform imputation. The paper empirically demonstrates the efficacy of the proposed NuwaTS framework on several benchmark time-series datasets, and domain-transfer capabilities that are crucial for a foundational model.

**Strengths:**

1. Imputation is an important step in almost every real-world time-series analysis, and designing a foundational model for general purpose time-series imputation is both interesting and useful for the community.
2. The proposed domain-specific embedding is novel, and allows for the lightweight (and plug-and-play) domain-specific fine-tuning of the NuwaTS model.

**Weaknesses:**

1. While the paper details a few time-series modeling approaches using pre-trained LMs in related works, the motivation for using them for imputation is missing. See additional details in questions.
2. The paper defines time-series imputation as a univariate problem. It leverages this formulation to introduce a novel benchmarking paradigm (called variable-wise division). However, multivariate time-series imputation has unique advantages over univariate formulations, such as the ability to leverage cross-variate correlations. See additional details in questions.
3. The proposed NuwaTS model has marginal improvements over the PatchTST (one-for-all) variant.

**Questions:**

1. Are there any specific advantages of using pre-trained LMs over standard transformer based time-series models such as PatchTST, iTransformer, Autoformer, Ti-MAE, etc.? How is pre-training on natural languages (using self-supervised learning) helpful for time-series imputation? Would randomly initialized LMs be able to achieve the same level of accuracy (i.e., training from scratch)?
2. Holistically comparing univariate time series imputation (using the variable-wise division paradigm) to the standard train/val/test splits in time (which allows for multivariate imputation) is important to demonstrate that univariate time-series imputation is sufficient. Imputation methods like SAITS, BRITS, etc. might be optimized for multivariate time-series imputation, and hence are not as good when compared using the proposed benchmarking paradigm.
3. The motivation behind using statistical embeddings is not clear. Will the performance of the NuwaTS significantly drop without this? Empirically demonstrating it using an ablation study could help highlight the importance.
4. Line 198: $Z_{i, (p)}$ notation needs clarification. $I$, $p$, $D$, and $N$ are not defined before.
5. Line 213: $z_{i, (p)}$ not defined. $r_i$ is not defined (I am guessing it is the mask ratio). Why is it multiplied with the mask ratio?
6. While the experiments on synthetic masking such as random masking and continuous masking are interesting and illustrative to an extent, adding more experiments on real-world datasets with missing values could strengthen the paper.

---

### Note · Authors · 2024-11-19

I have read and agree with the venue's withdrawal policy on behalf of myself and my co-authors.